# Association between continuous glucose monitoring-derived metrics and coronary plaque vulnerability: A retrospective exploratory analysis

Hikaru Sugimoto[1], Ken-ichi Hironaka[2], Tomoko Yamada[3], Natsu Otowa-Suematsu[3], Yushi Hirota[3], Hiromasa Otake[4], Ken-Ichi Hirata[3], Kazuhiko Sakaguchi[3], Wataru Ogawa[3]*, Shinya Kuroda[1,2]*

[1]Department of Biochemistry and Molecular Biology, Graduate School of Medicine, The University of Tokyo, Tokyo, Japan; [2]Department of Biological Sciences, Graduate School of Science, The University of Tokyo, Tokyo, Japan; [3]Division of Diabetes and Endocrinology, Department of Internal Medicine, Kobe University Graduate School of Medicine, Kobe, Japan; [4]Division of Cardiovascular Medicine, Department of Internal Medicine, Kobe University Graduate School of Medicine, Kobe, Japan

*For correspondence:
ogawa@med.kobe-u.ac.jp (WO);
skuroda@bs.s.u-tokyo.ac.jp (SK)

**Competing interest:** The authors declare that no competing interests exist.

## eLife Assessment

This **valuable** retrospective analysis identified three independent components of glucose dynamics - "value", "variability", and "autocorrelation" - which may be used in predicting coronary plaque vulnerability. The study is **solid** and of interest to a wide range of investigators in the medical field who are interested in the role of glycemia on cardiometabolic health. The manuscript has been substantially strengthened by clarifying methods, improving transparency, and validating key findings, resulting in a coherent and persuasive case for autocorrelation as a meaningful third dimension of glucose dynamics despite remaining design-related limitations.

## Abstract

**Background:** Impaired glucose homeostasis leads to numerous complications, with coronary artery disease (CAD) being a major contributor to healthcare costs worldwide. Because continuous glucose monitoring (CGM) captures multidimensional features of glucose regulation beyond average glycemia, we evaluated whether CGM-derived indices better predict coronary plaque vulnerability than conventional measures.

**Methods:** We examined associations between CGM-derived indices and coronary plaque vulnerability assessed by virtual histology–intravascular ultrasound, focusing on the necrotic core (%NC) in humans. We analyzed 14 CGM-derived indices, including average daily risk ratio (ADRR) and autocorrelation-based metrics (AC_Mean and AC_Var), alongside commonly used measures, such as fasting blood glucose (FBG), hemoglobin A1c (HbA1c), and 120 min plasma glucose during oral glucose tolerance testing (PG120). Factor analysis was used to identify latent components underlying glucose dynamics and to relate these components to %NC. Findings were validated across independent datasets from Japan (n=64), the United States (n=53), and China (n=100).

**Results:** CGM-derived indices, particularly ADRR and AC_Var, demonstrated stronger predictive capability for %NC than FBG, HbA1c, and PG120. Factor analysis identified three independent components of glucose dynamics: mean, variance, and autocorrelation, each showing an independent association with %NC. ADRR reflected both mean and variance components, whereas AC_Var

primarily captured the autocorrelation component. In contrast, FBG, HbA1c, and PG120 primarily reflected the mean component alone and were, therefore, insufficient for %NC prediction.

**Conclusions:** CGM-derived indices reflecting the three components of glucose dynamics can serve as more effective screening tools for CAD risk assessment, complementing or possibly replacing traditional diabetes diagnostic methods.

**Funding:** This study was supported by the Japan Society for the Promotion of Science (JSPS) KAKENHI (JP21H04759), CREST, the Japan Science and Technology Agency (JST) (JPMJCR2123), The Uehara Memorial Foundation, and The Takeda Science Foundation.

## Introduction

Diabetes mellitus (DM) affects more than 400 million people globally, and coronary artery disease (CAD) is a major driver of morbidity, mortality, and healthcare costs among individuals with type 2 DM (T2DM) (*Bax et al., 2007*). Various prognostic models (*Fiarni et al., 2019*; *Ravaut et al., 2021*) and diagnostic markers (*Bax et al., 2007*) for CAD have been developed; however, population-level screening remains imperfect, often being ineffective, costly, or labor-intensive (*Bax et al., 2007*; *Young et al., 2009*). There is a need for scalable approaches that can identify individuals at the highest risk using easily accessible clinical information.

Blood glucose levels are among the readily obtained predictors of the complications (*Psoma et al., 2024*). The disrupted conditions of glucose dynamics seen in impaired glucose tolerance (IGT) and T2DM are partly characterized by high concentrations of blood glucose levels (*Monnier et al., 2008*). High concentrations of blood glucose levels have been defined as having high hemoglobin A1c (HbA1c) levels, fasting blood glucose (FBG) levels, and plasma glucose concentration at 120 min during the oral glucose tolerance test (OGTT) (PG120) (*Monnier et al., 2008*). These indices, especially HbA1c, are associated with complications of T2DM (*Selvin et al., 2010*).

Recent studies have shown that glucose variability, in addition to absolute glucose concentration, significantly contributes to the prognosis of complications (*Gerbaud et al., 2019*; *Gorst et al., 2015*; *Monnier et al., 2008*; *Psoma et al., 2024*; *Su et al., 2011*; *Zhou et al., 2018*) and all-cause mortality (*Cai et al., 2023*). Continuous glucose monitoring (CGM) can estimate short-term glycemic variability (*Service, 2013*) and has been reported to predict T2DM complications (*Tang et al., 2016*). Standard deviation (Std) of glucose levels (CGM_Std), mean amplitude of glycemic excursion (MAGE), mean of daily difference (MODD), and continuous overlapping net glycemic action (CONGA) are established indices of glycemic variability, of which CGM_Std and MAGE are more highly correlated with coronary plaque properties (*Otowa-Suematsu et al., 2018*). Among glucose level-related indices, including HbA1c and FBG, MAGE is an independent determinant of coronary plaque instability (*Okada et al., 2015*).

Other CGM-derived indices, such as average daily risk ratio (ADRR), lability index (LI), J-index, mean absolute glucose (MAG), and glycemic risk assessment in diabetes equation (GRADE) have also been developed (*Hill et al., 2011*). We recently showed that AC_Mean and AC_Var, which are calculated from the autocorrelation function of glucose levels measured by CGM, can detect decreased abilities in glucose regulation that cannot be captured by FBG, HbA1c, or the other conventional CGM-derived indices (*Sugimoto et al., 2025*). The characteristics of glucose dynamics can also be estimated from insulin concentrations. The disposition index (DI), which is the product of insulin sensitivity and insulin secretion, reflects and predicts glycemic disability beyond FBG (*Utzschneider et al., 2009*). Several other glucose-related indices and the relationship between the indices have also been reported (*Fabris et al., 2015*; *Fabris et al., 2014*; *Keshet et al., 2023*). Despite these advances, a comprehensive understanding of how these various indices can be optimally combined to predict T2DM complications, particularly CAD, remains elusive. Furthermore, the underlying factors that these indices represent and their individual associations with CAD remain to be fully elucidated.

This study aims to address these knowledge gaps through three objectives: (i) to determine which clinical parameters are effective predictors of coronary plaque vulnerability; (ii) to identify the factors underlying these indices; and (iii) to elucidate how these factors are associated with coronary plaque vulnerability. We investigated the characteristics of 14 CGM-derived indices: 12 relatively well-known CGM-derived indices (*Hill et al., 2011*) and 2 indices (AC_Mean and AC_Var), as well as OGTT-derived indices, and investigated the relationship between these parameters and coronary plaque

vulnerability assessed by virtual histology-intravascular ultrasound (VH-IVUS). We showed that three factors, namely, mean, variance, and autocorrelation, underlie blood glucose level-related indices, and that the three are independently associated with coronary plaque vulnerability.

## Results

### Mean, standard deviation, and autocorrelation function of glucose levels independently contribute to the prediction of coronary plaque vulnerability

We previously showed that AC_Var, a measure derived from the autocorrelation function of CGM time series, detects impaired glucose handling that is not reflected by the CGM-derived mean (CGM_Mean) or standard deviation (CGM_Std) (*Sugimoto et al., 2025*). Based on the study, we hypothesized that AC_Var could identify individuals with high %NC, a widely used parameter of plaque vulnerability, independently from CGM_Mean and CGM_Std. To test this hypothesis, we conducted a multiple regression analysis with CGM_Mean, CGM_Std, and AC_Var as input variables and %NC as the objective variable using a previously described dataset consisting of 8 individuals with normal glucose tolerance (NGT), 16 with IGT, and 29 with T2DM (*Otowa-Suematsu et al., 2018*).

The variance inflation factors (VIFs) for CGM_Mean, CGM_Std, and AC_Var were 1.1, 1.1, and 1.0, respectively, indicating low multicollinearity among these variables. The regression model including CGM_Mean, CGM_Std, and AC_Var to predict %NC achieved an $R^2$ of 0.36 and an Akaike Information Criterion (AIC) of 321. Each of these indices showed a statistically significant independent positive correlation with %NC (*Figure 1A*). In contrast, the model using conventional glycemic markers (FBG, HbA1c, and PG120) yielded an $R^2$ of only 0.05 and an AIC of 340 (*Figure 1B*). Similarly, the model using the Framingham Risk Score for Hard Coronary Heart Disease (*Wilson et al., 1998*) showed limited predictive value, with an $R^2$ of 0.04 and an AIC of 330 (*Figure 1C*). The AC_Var computed from 15 min CGM sampling was nearly identical to that computed from 5 min sampling (*R*=0.99, 95% CI: 0.97–1.00) (*Figure 1—figure supplement 1A*), and the regression using the 15 min features yielded almost the same performance ($R^2$=0.36; AIC = 320; *Figure 1—figure supplement 1B*). Collectively, these results indicate that CGM_Mean, CGM_Std, and AC_Var provide complementary and independent information for predicting coronary plaque vulnerability.

### Association among clinical variables

To further characterize CGM-derived indices in estimating %NC, we examined Spearman's correlation coefficients (*r*) between CGM-derived indices and %NC (*Figure 1D and E*). For comparison, we also investigated FBG, HbA1c, OGTT-derived indices, body mass index (BMI), triglycerides (TGs), low-density lipoprotein cholesterol (LDL-C), high-density lipoprotein cholesterol (HDL-C), systolic blood pressure (SBP), and diastolic BP (DBP). Given that this study enrolled individuals with well-controlled serum cholesterol and BP levels (*Figure 1—figure supplement 1C*), weak correlations between these indices and %NC were expected.

A correlation network connecting relationships with Q<0.05 showed that AC_Var was statistically significantly correlated with %NC (*r*=0.35; 95% CI, 0.09–0.57) (*Figure 1D and E*). By contrast, AC_Var showed relatively weak correlations with other indices, including CGM_Mean (*r* = –0.02; 95% CI, –0.30–0.24) and CGM_Std (*r*=0.15; 95% CI, –0.14–0.43) (*Figure 1F*). Twelve CGM-derived indices, ADRR, MAGE, JINDEX, CGM_Std, CGM_Mean, GRADE, MVALUE, AC_Var, LI, HBGI, CONGA, and MODD, showed significant correlations with %NC (*Figure 1E*, *Figure 1—figure supplement 1D*). By contrast, with the exception of the insulinogenic index (I.I.), OGTT-derived indices, FBG, and HbA1c showed relatively weak correlations with %NC.

To assess the multicollinearity among the input variables, we examined the VIF for each variable (*Figure 1G and H*). We removed variables with the highest VIF one by one until all variables had VIF values less than 10. This procedure retained 19 variables when considering the full set of clinical and CGM measures (*Figure 1G*) and eight variables when considering only CGM-derived measures (*Figure 1H*). AC_Var was retained in both sets, indicating relatively low multicollinearity with the other indices.

To examine the reproducibility of the relationship among the input variables, we compared VIF values calculated from this dataset (VIFt) with those from previously reported datasets (VIFp1

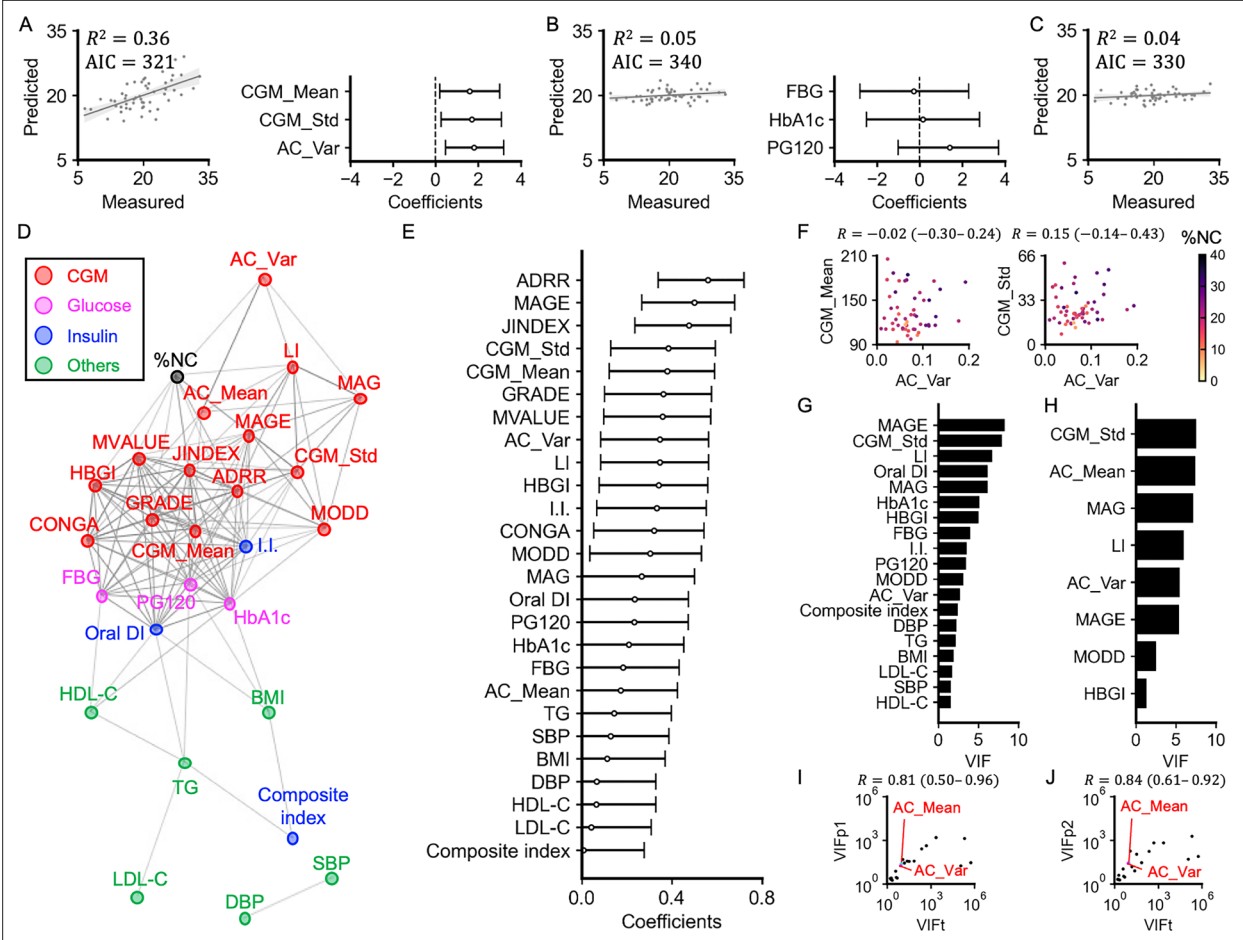

**Figure 1.** Relations among clinical measures. (**A**) Multiple regression analysis between necrotic core (%NC) and continuous glucose monitoring (CGM)_Mean, CGM_Std, and autocorrelation (AC)_Var. Scatter plots for predicted %NC versus measured %NC (the left). Each point corresponds to the values for a single individual. Bars represent the 95% confidence intervals (CIs) of the regression coefficients (the right). (**B**) Multiple regression analysis between %NC and fasting blood glucose (FBG), HbA1c, and PG120. (**C**) Linear regression analysis between %NC and the Framingham Risk Score for Hard Coronary Heart Disease. (**D**) A spring layout of the correlation network involving %NC (black), 14 CGM-derived indices (red), three blood glucose level-related indices (magenta), three insulin sensitivity or secretion-related indices (blue), and six other clinical indices (green) obtained from a single blood test or physical measurement. Connections denote relationships with Q<0.05. The width of the edges is proportional to the corresponding correlation coefficient. (**E**) The absolute values of Spearman's correlation coefficients between clinical parameters and %NC. Bars represent the 95% CIs. (**F**) Scatter plots for AC_Var versus CGM_Mean (the left), and AC_Var versus CGM_Std (the right). Each point corresponds to a single individual's values. Individuals are colored based on their %NC value. *R* is Spearman's correlation coefficient, and the value in parentheses is 95% CI. (**G, H**) Variance inflation factor (VIF) of each variable remaining after removing the variables with the highest VIF one by one until the VIFs of all variables are less than 10. The input variables include the following 26 variables: BMI, SBP, DBP, TG, LDL-C, HDL-C, FBG, HbA1c, PG120, I.I., composite index, Oral disposition index (DI), CGM_Mean, CGM_Std, CONGA, LI, JINDEX, HBGI, GRADE, MODD, MAGE, ADRR, MVALUE, MAG, AC_Mean, and AC_Var. The input variables of (**H**) included only the CGM-derived measures from the above 26 indices. (**I**) Scatter plot of the VIF of the indices measured in this study (VIFt) versus that of indices measured in a previous study (VIFp1) (*Sugimoto et al., 2025*). R is Spearman's correlation coefficient, and the value in parentheses is the 95% CI. (**J**) Scatter plot of the VIF of the indices measured in this study (VIFt) versus that of the indices measured in a previous study (*Hall et al., 2018*) (VIFp2).

The online version of this article includes the following figure supplement(s) for figure 1:

**Figure supplement 1.** Relationship among clinical parameters.

*Sugimoto et al., 2025* and VIFp2 *Hall et al., 2018*). VIFt significantly correlated with both VIFp1 and VIFp2 (*Figure 1I and J*). VIFs for AC_Mean and AC_Var were consistently low across the datasets, suggesting the reproducibility of our observation that AC_Mean and AC_Var show relatively low multicollinearity with other CGM-derived indices.

## CGM-derived indices, particularly ADRR, AC_Var, MAGE, and LI, contribute to the prediction of coronary plaque vulnerability

To investigate which variables are particularly useful in estimating %NC, we used two statistical techniques: Least Absolute Shrinkage and Selection Operator (LASSO) regression and Partial Least Squares (PLS) regression (*Figure 2*; *Tibshirani, 1996*; *Wold et al., 2001*). These regression models have been used for studies where the number of input variables is large relative to the sample size (*Pei et al., 2023*; *Wang et al., 2005*).

LASSO uses L1 regularization to produce models with fewer parameters and has been widely used for feature selection in predictive modeling (*Wei et al., 2022*). We included BMI, FBG, HbA1c, OGTT-derived indices, and CGM-derived indices as the input variables. The leave-one-out cross-validation identified the optimal regularization coefficient, lambda, as 0.849 (*Figure 2A*). At the lambda, the coefficients of ADRR, AC_Var, MAGE, and LI were estimated to be non-zero coefficients (*Figure 2B and C*), suggesting that CGM-derived indices, particularly ADRR, AC_Var, MAGE, and LI, contribute to the prediction of %NC. Even with the inclusion of SBP, DBP, TG, LDL-C, and HDL-C as additional input variables, the results remained consistent, with the coefficients of ADRR, AC_Var, MAGE, and LI still estimated as non-zero coefficients (*Figure 2—figure supplement 1A–C*).

To further validate the LASSO results and address potential instability, we performed PLS regression and examined the Variable Importance in Projection (VIP) scores (*Figure 2D*). PLS regression is particularly useful when dealing with many input variables that may be highly collinear (*Wold et al., 2001*). The VIP scores of ADRR, AC_Var, MAGE, and LI, which were estimated to be non-zero coefficients by LASSO, were higher than 1, indicating that these four variables especially contribute to the prediction of %NC. Conventional indices, including FBG, HbA1c, PG120, I.I., Composite index, and Oral DI, did not contribute significantly to the prediction compared to these CGM-derived indices. Even when SBP, DBP, TG, LDL-C, and HDL-C were included as additional input variables, the results remained consistent, and the VIP scores for ADRR, AC_Var, MAGE, and LI remained greater than 1 (*Figure 2—figure supplement 1D*).

## Three components of glucose dynamics – mean, variance, and autocorrelation – are associated with coronary plaque vulnerability

To elucidate the underlying factors of clinical parameters and their association with %NC, we performed an exploratory factor analysis. Factor analysis reduces interrelated indices into a smaller set of underlying common factors and has been employed to examine the interdependencies among various clinical parameters (*Augstein et al., 2015*; *Cappelleri et al., 2000*; *Oh et al., 2004*) and DM complications (*Guo et al., 2020*).

The optimal number of underlying factors was determined using Bayesian information criterion (BIC) and minimum average partial (MAP) methods, which indicated that five and six factors were appropriate, respectively. We first set the number of underlying factors as five. *Figure 3A* shows that FBG, HbA1c, PG120, I.I., oral DI, CGM_Mean, CONGA, HBGI, MVALUE, GRADE, JINDEX, and ADRR were included in the first factor. Given that most of these indices are related to the value of blood glucose concentration, factor 1 was labeled 'mean.' CGM_Std, MAGE, LI, MAG, MODD, JINDEX, and ADRR were included in the second factor. Given that these indices are related to glucose variability, factor 2 was labeled 'variance.' Given that the definition of JINDEX is based on the sum of CGM_Mean and CGM_Std, and that of ADRR is based on both high and low values of glucose, the result that JINDEX and ADRR clustered in both factors 1 and 2 is plausible. Given that autocorrelation-derived indices, AC_Mean and AC_Var, were included in the third factor, factor 3 was labeled 'autocorrelation.' BMI, PG120, composite index, and oral DI were included in the fourth factor. Factor 4 did not include any CGM-derived indices. Given that this combination of indices indicates a decrease in oral DI and associated increase in blood glucose due to decreased insulin sensitivity, factor 4 was labeled 'sensitivity (without CGM).' PG120, I.I., oral DI, and MAG were included in the fifth factor. Factor 5 did not have positive loadings of any CGM-derived indices. Given that this combination of the indices indicates a decrease in oral DI and associated increase in blood glucose due to decreased insulin secretion (I.I.), factor 5 was labeled 'secretion (without CGM).' Each factor explained 39%, 21%, 10%, 5%, and 5% of the total variance of the factors, respectively.

The validity of the factor analysis was assessed according to previous studies (*Cappelleri et al., 2000*; *Guo et al., 2020*). To evaluate the applicability of the factor analysis, the Kaiser-Meyer-Olkin

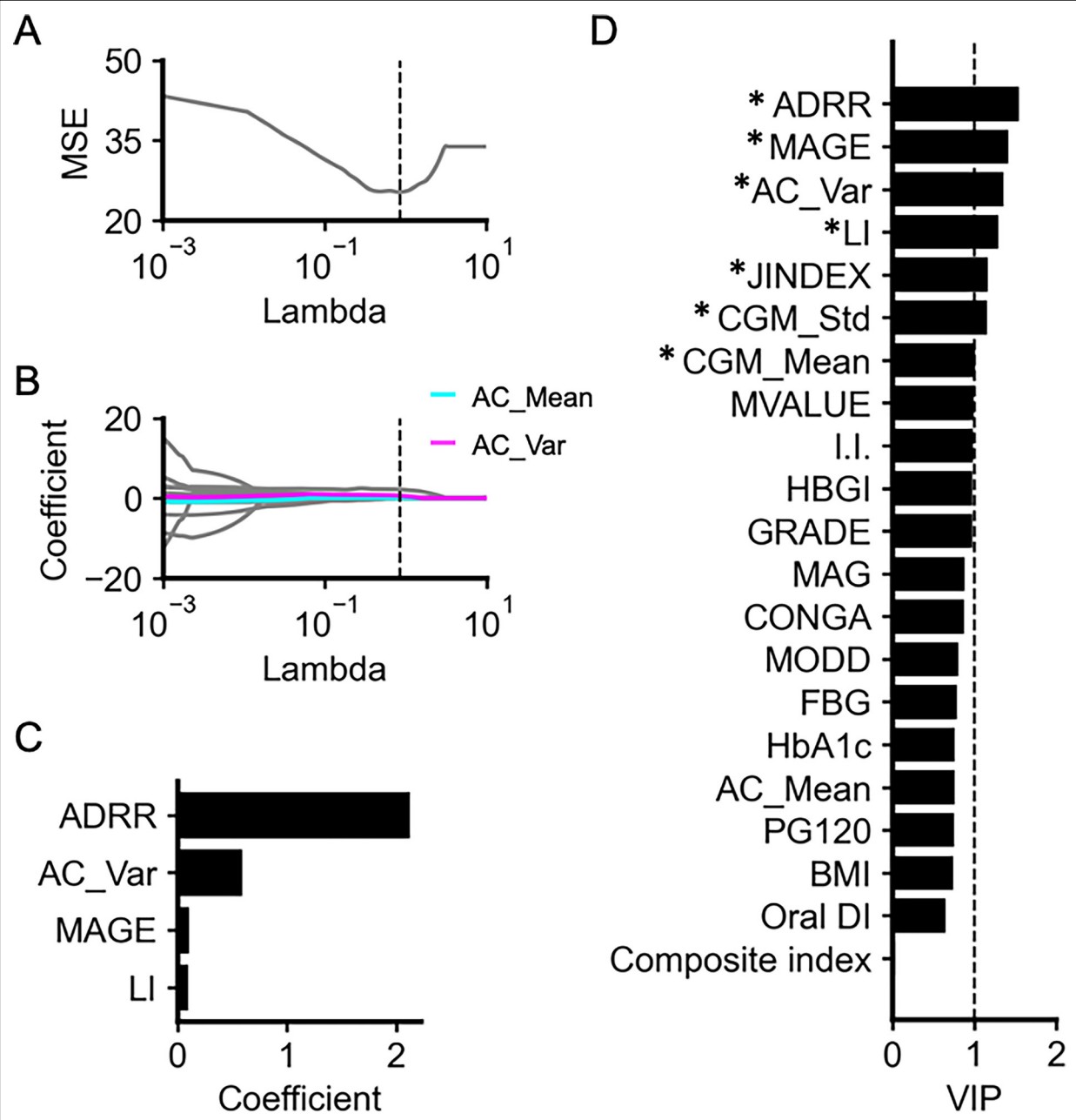

**Figure 2.** Least Absolute Shrinkage and Selection Operator (LASSO) and Partial Least Squares (PLS) regression analyses for predicting necrotic core (%NC). (**A**) Relationship between regularization coefficients (lambda) and the mean-squared error (MSE) based on the leave-one-out cross-validation in predicting %NC. A dotted vertical line indicates the optimal lambda, which provides the lowest MSE. The optimal lambda was 0.849. (**B**) LASSO regularization paths along the lambda in predicting %NC. Cyan, magenta, and gray lines indicate the estimated coefficients of autocorrelation (AC)_Mean, AC_Var, and the other input variables, respectively. A dotted vertical line indicates the optimal lambda. (**C**) Estimated coefficients with the optimal lambda. Only variables with non-zero coefficients are shown. Input variables include the following 21 variables: body mass index (BMI), fasting blood glucose (FBG), HbA1c, PG120, I.I., Composite index, Oral DI, continuous glucose monitoring (CGM)_Mean, CGM_Std, CONGA, LI, JINDEX, HBGI, GRADE, MODD, MAGE, ADRR, MVALUE, MAG, AC_Mean, and AC_Var. (**D**) Variable Importance in Projection (VIP) generated from the PLS regression predicting %NC. Variables with a VIP ≥1 (the dotted line) were considered to significantly contribute to the prediction.

The online version of this article includes the following figure supplement(s) for figure 2:

**Figure supplement 1.** Least Absolute Shrinkage and Selection Operator (LASSO) and Partial Least Squares (PLS) regression analyses for predicting necrotic core (%NC) including systolic blood pressure (SBP), diastolic blood pressure (DBP), triglycerides (TGs), low-density lipoprotein cholesterol (LDL-C), and high-density lipoprotein cholesterol (HDL-C).

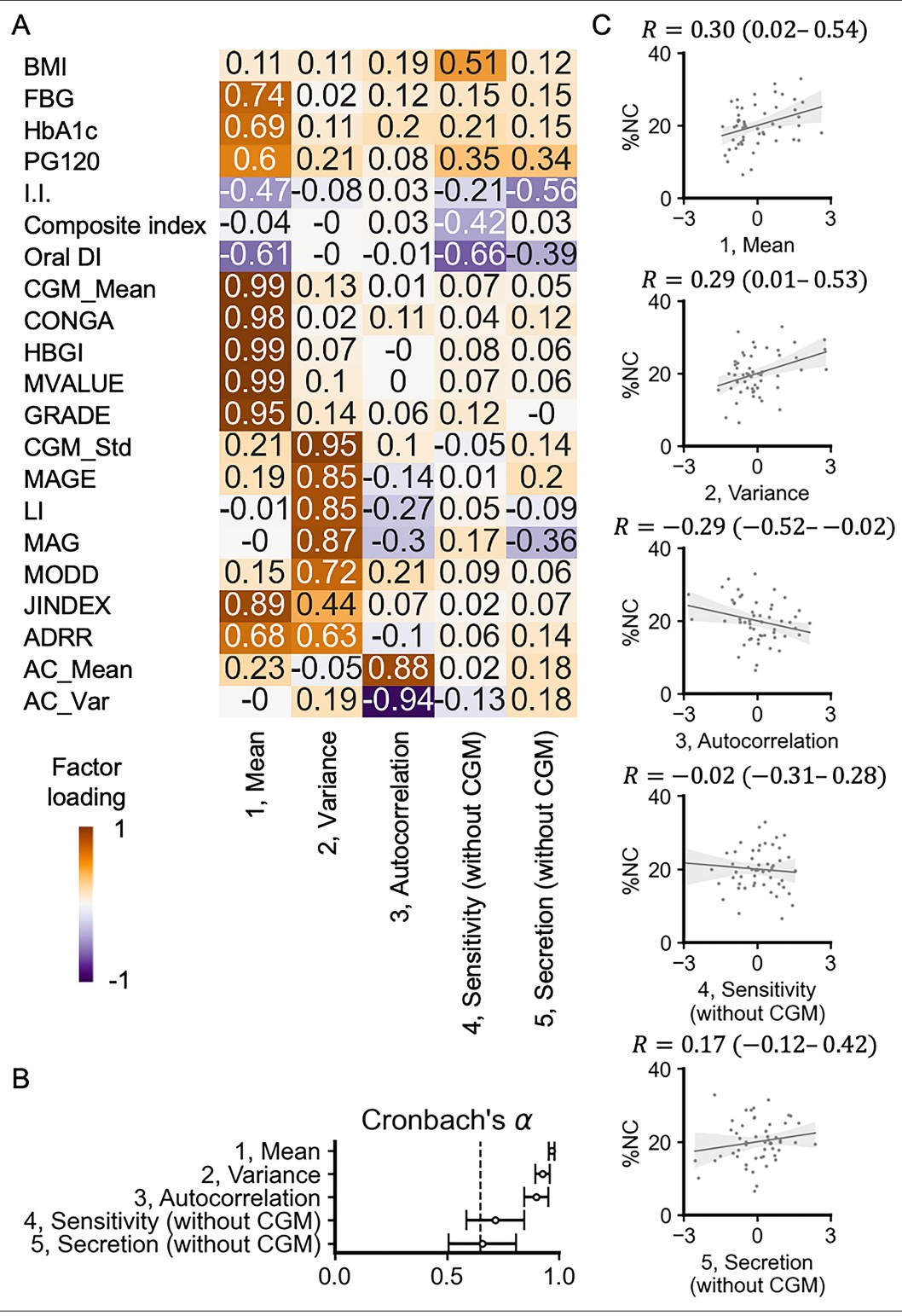

**Figure 3.** Factor analysis of the clinical parameters. (**A**) Factor analysis after orthogonal rotation. The values and colors are based on the factor loadings. The columns represent each factor. The rows represent input indices. (**B**) Cronbach's α for each factor. Bars represent the 95% confidence interval (CI). (**C**) Scatter plots and fitted linear regression lines for factor scores versus necrotic core (%NC). Each point corresponds to the values for a single individual. *R* is Spearman's correlation coefficient, and the value in parentheses is the 95% CI.

The online version of this article includes the following figure supplement(s) for figure 3:

*Figure 3 continued on next page*

*Figure 3 continued*

**Figure supplement 1.** Factor analyses of the clinical parameters.

**Figure supplement 2.** Hierarchical clustering analysis of metabolic syndrome-related indices.

**Figure supplement 3.** Factor analyses of the clinical parameters using the previously reported datasets.

**Figure supplement 4.** Factor analysis of continuous glucose monitoring (CGM)-derived indices using a previously reported dataset.

(KMO) and Bartlett's spherical test were performed. The KMO test indicated that the value of the measure of sampling adequacy for this data was 0.64, and Bartlett's spherical test indicated that the variables were statistically significantly intercorrelated ($p<0.01$), suggesting that this dataset was applicable for the factor analysis. To evaluate the internal consistency, Cronbach's α (*Figure 3B*) and item–total correlations were calculated for each factor. Cronbach's α was 0.97 for factor 1, 0.93 for factor 2, 0.90 for factor 3, 0.72 for factor 4, and 0.66 for factor 5; these values were larger than 0.65, suggesting that the internal consistency was satisfactory. While Cronbach's α of factor 5 was relatively low, exclusion of MAG increased the Cronbach's α to 0.84, indicating that the association between factor 5 and decrease in oral DI and associated increase in blood glucose due to decreased insulin secretion could be considered reliable. Item–total correlations ranged from 0.63 to 0.97 for factor 1, 0.72–0.94 for factor 2, 0.82 for factor 3, 0.54–0.76 for factor 4, and 0.37–0.86 for factor 5. With the exception of MAG, item–total correlations ranged from 0.84 to 0.91 for factor 5. The correlation coefficient of MAG was 0.37, which can be considered a modest correlation (*Cappelleri et al., 2000*), and the item–total correlations were generally reasonably strong in demonstrating reliability.

We also investigated a 6-factor solution (*Figure 3—figure supplement 1A*). Factors 1, 2, and 3 could be interpreted as mean, variance, and autocorrelation, respectively, similar to the 5-factor solution. Given that factor 6 had no factor loadings ≥0.3, we applied the 5-factor solution in the subsequent analysis. Furthermore, the inclusion of SBP, DBP, TG, LDL-C, and HDL-C into the input variables did not change the presence of the three components (mean, variance, and autocorrelation) in glucose dynamics (*Figure 3—figure supplement 1B*). Since we included only individuals with well-controlled serum cholesterol and BP levels in this study, we applied the 5-factor solution without these indices (*Figure 3A*) to the following analysis.

To further examine the stability of the results of the factor analysis, we also conducted hierarchical clustering analysis (*Figure 3—figure supplement 2*). The optimal number of clusters was determined based on silhouette analysis. A large positive silhouette coefficient indicates that each cluster is compact and distinct from the others. The analysis indicated that the four clusters were appropriate (*Figure 3—figure supplement 2A*). Clusters 1, 2, and 3 can be interpreted as mean, variance, and autocorrelation, respectively (*Figure 3—figure supplement 2B*).

To investigate the association between these underlying factors and %NC, we investigated the correlation between the factor scores and %NC (*Figure 3C*). The factor mean and variance showed significant positive correlations with %NC, whereas autocorrelation showed a significant negative correlation. Factors 4 and 5, which were less related to the CGM-derived indices, showed weaker correlations with %NC. Collectively, we conclude that glucose dynamics have three components – mean, variance, and autocorrelation – and that these three components are associated with %NC.

To assess the robustness and generalizability of our factor analysis results, we performed similar analyses using previously published datasets from diverse populations (*Figure 3—figure supplements 3 and 4*). Factors that could be interpreted as representing the mean, variance, and autocorrelation of glucose dynamics were consistently observed across diverse populations, including Japanese (*Sugimoto et al., 2025*; *Figure 3—figure supplement 3A*), American (*Hall et al., 2018*; *Figure 3—figure supplement 3B*), and Chinese (*Zhao et al., 2023*; *Figure 3—figure supplement 4A*) datasets. In the Chinese dataset, all three factors were significantly different between individuals with and without diabetic macrovascular complications (*Figure 3—figure supplement 4B*). Taken together, these results support the reproducibility and cross-cultural validity of our three-component model of glucose dynamics.

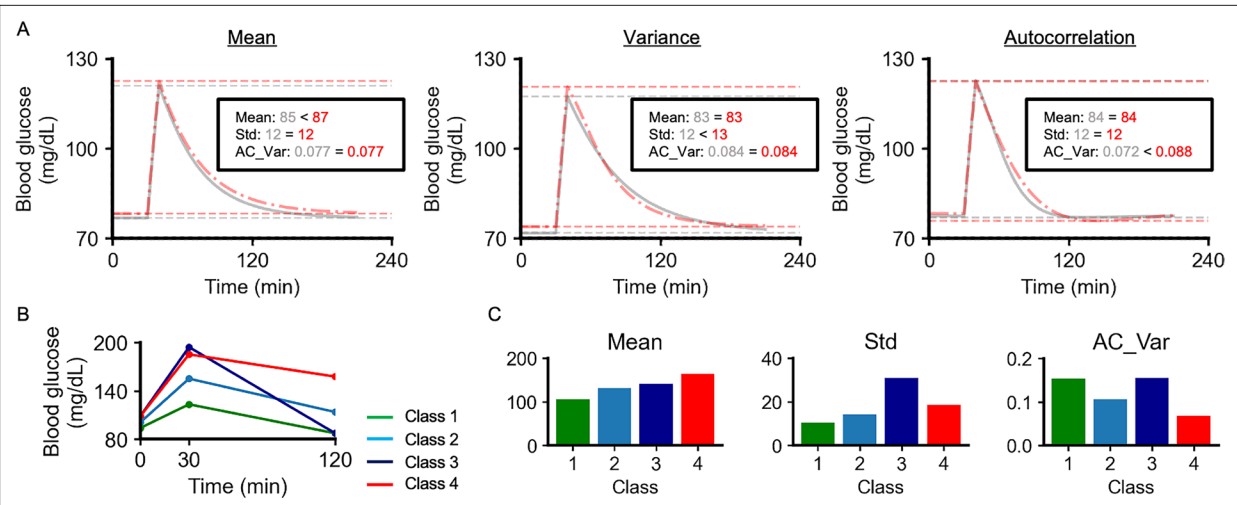

**Figure 4.** Overview of the three components of glucose dynamics. (**A**) 240 min simulated glucose concentration. The colors of the lines are based on the mean (Mean), Std, and autocorrelation (AC)_Var of the simulated blood glucose. Red and gray dotted horizontal lines indicate the minimum and maximum blood glucose values, respectively. (**B**) Previously reported patterns of blood glucose during the oral glucose tolerance test (OGTT) (*Hulman et al., 2018*). Green, class 1; light blue, class 2; dark blue, class 3; red, class 4. (**C**) Mean, Std, and AC_Var of the glucose during the OGTT. Colors are based on the classes shown in (**B**).

The online version of this article includes the following figure supplement(s) for figure 4:

**Figure supplement 1.** Infused glucose and parameters used in simulating glucose dynamics.

**Figure supplement 2.** Relation between continuous glucose monitoring (CGM)-derived measures and clinical measures.

## Overview of the three components of glucose dynamics – mean, variance, and autocorrelation

We have shown that three components of glucose dynamics – mean, variance, and autocorrelation – are associated with %NC. To overview the characteristics of glucose dynamics with different values of the components, we simulated glucose fluctuations using a previously reported mathematical model (*De Gaetano and Arino, 2000*; *Figure 4*, *Figure 4—figure supplement 1*). We could generate glucose fluctuations with almost the same standard deviation (Std) and AC_Var but with a different mean (*Figure 4A*). Similarly, we could also simulate glucose fluctuations with almost the same mean and AC_Var but different Std, and with almost the same mean and Std but different AC_Var. These three components were independently adjustable by changing parameters within the range of values for NGT individuals (*Figure 4—figure supplement 1B*). Individuals with higher AC_Var tended to have higher %NC (*Figure 1*); however, comparing the glucose dynamics with higher and lower AC_Var, the maximum value of blood glucose was lower in individuals with higher AC_Var (*Figure 4A*).

We then investigated the relationship between the three components and shapes of the glucose response curve after OGTT. Patterns of the glucose response curve after OGTT were heterogeneous, and four distinct patterns, denoted classes 1–4, were previously identified by analyzing the glucose dynamics of 5861 individuals in Denmark (*Figure 4B*; *Hulman et al., 2018*). The classes with a high mean did not necessarily have high Std or AC_Var (*Figure 4C*). The classes with high Std did not necessarily have high mean and AC_Var (*Figure 4C*), consistent with the result that mean, Std, and AC_Var had low multicollinearity with each other. Compared to class 2, only class 1 was lower in mean and Std and was higher in AC_Var. Compared to class 2, only class 3 was higher in mean, Std, and AC_Var. Compared to class 2, only class 4 was higher in mean and Std and lower in AC_Var. Collectively, the three components could characterize the previously reported four distinct patterns during OGTT. Class 3 was characterized by normal FBG and PG120 values but has been associated with increased risk of diabetes and a higher all-cause mortality rate, suggesting that subgroups at high risk may not be identified by investigating only FBG and PG120 (*Hulman et al., 2018*). Std and AC_Var were high in class 3 (*Figure 4C*), suggesting that high Std and high AC_Var indicate glycemic disability independent of FBG and PG120.

## Discussion

Our results show that glucose dynamics consist of three distinct components - mean, variance, and autocorrelation - each of which is independently associated with coronary plaque vulnerability. We further show that traditional diabetes diagnostics rely on indices, such as FBG, PG120, and HbA1c that primarily reflect only the mean component, and that incorporating all three components significantly improves the prediction of coronary plaque vulnerability. This result is partially consistent with a previous notion that glucose dynamics include two components: amplitude and timing (*Cobelli and Facchinetti, 2018*). Given our previous study demonstrating that AC_Var, an index reflecting autocorrelation, identifies impaired glucose regulation independently of CGM-derived indices representing mean and variance (*Sugimoto et al., 2025*), CGM-derived indices incorporating all three components may outperform conventional diabetes diagnostics in predicting both glycemic control capacity and coronary plaque vulnerability.

We also showed that CGM-derived indices, especially ADRR and AC_Var, contribute to the prediction of %NC by using LASSO and PLS. Given that the definition equation for ADRR is affected by both high and low concentrations of blood glucose (*Hill et al., 2011*), it is likely affected by both the mean and variance components. Factor analysis confirmed that ADRR was included in both factor 1 (mean) and factor 2 (variance). Since three factors, mean, variance, and autocorrelation, contribute independently to the prediction of the complication, it would be useful to examine ADRR, which is influenced by both mean and variance, and AC_Var, which is influenced by autocorrelation, in predicting %NC with a minimal number of variables.

This study also provided evidence that autocorrelation can vary independently from the mean and variance components using simulated data. In addition, simulated glucose dynamics indicated that even individuals with high AC_Var did not necessarily have high maximum and minimum blood glucose levels. This study also indicated that these three components qualitatively corresponded to the four distinct glucose patterns observed after glucose administration, which were identified in a previous study (*Hulman et al., 2018*). Thus, the inclusion of autocorrelation in addition to mean and variance may improve the characterization of inter-individual differences in glucose regulation and improve the predictive accuracy of various clinical outcomes.

Despite increasing evidence linking glycemic variability to oxidative stress and endothelial dysfunction in T2DM complications (*Ceriello et al., 2008*; *Monnier et al., 2008*), the biological mechanisms underlying the independent predictive value of autocorrelation remain to be elucidated. Our previous work has shown that glucose autocorrelation is influenced by insulin clearance (*Sugimoto et al., 2025*), a process known to be associated with cardiovascular disease risk (*Randrianarisoa et al., 2020*). Therefore, the molecular pathways linking glucose autocorrelation to cardiovascular disease may share common mechanisms with those linking insulin clearance to cardiovascular disease. Although previous studies have primarily focused on investigating the molecular mechanisms associated with mean glucose levels and glycemic variability, our findings open new avenues for exploring the molecular basis of glucose autocorrelation, potentially revealing novel therapeutic targets for preventing diabetic complications.

The current study had several limitations. Although our LASSO and factor analysis indicated that CGM-derived measures were strong predictors of %NC, this does not mean that other clinical parameters, such as lipids and blood pressure, are irrelevant in T2DM complications. Our study specifically focused on characterizing glucose dynamics, and we analyzed individuals with well-controlled serum cholesterol and blood pressure to reduce confounding effects. While we anticipate that inclusion of a more diverse population would not alter our primary findings regarding glucose dynamics, it is likely that a broader data set would reveal additional predictive contributions from lipid and blood pressure parameters. In our analysis of demographic factors, we found that age and gender had minimal influence on glucose dynamics characteristics (*Figure 4—figure supplement 2A, B*), suggesting that our findings regarding the relationship between glucose dynamics and coronary risk are robust across different demographic groups within our dataset. Future studies involving larger and more diverse populations would be valuable to comprehensively elucidate the potential influence of age, gender, and other demographic factors on glucose dynamics characteristics and their relationship to cardiovascular risk.

A previous study identified components of interday variability and hypoglycemia in CGM-derived indices (*Augstein et al., 2015*) that were not observed in our analysis. This discrepancy may be

due to the relatively small number of T2DM individuals in our study. We acknowledge that factor analyses of data from longer measurement periods, including more patients with T1DM and T2DM, could potentially yield different results. However, it is noteworthy that our analysis of longer-term CGM data sets from Japanese and American populations confirmed the existence of the same three factors - mean, variance, and autocorrelation - in glucose dynamics. Moreover, even with the short measurement period, CGM-derived indices reflecting these three factors demonstrated superior predictive accuracy for %NC compared to traditional indices, such as FBG, HbA1c, and PG120, underscoring the potential utility of CGM. Although time in range (TIR) was not included in the main analyses due to the relatively small number of T2DM patients and the predominance of participants with TIR >70%, our results demonstrate that CGM-derived indices outperformed conventional markers, such as FBG, HbA1c, and PG120 in predicting %NC. Furthermore, multiple regression analysis between factor scores and TIR revealed that only factor 1 (mean) and factor 2 (variance) were significantly associated with TIR (*Figure 4—figure supplement 2C, D*). This finding confirms the presence of three distinct components in glucose dynamics and highlights the added value of examining AC_Var as an independent glycemic feature beyond conventional CGM-derived measures. Some indices, such as HBGI, showed variation in classification across datasets, with some populations showing higher factor loadings in the 'mean' component and others in the 'variance' component. This variation occurs because HBGI calculations depend on the number of glucose readings above a threshold. In populations where mean glucose levels are predominantly below this threshold, the HBGI is more sensitive to glucose variability (*Figure 3—figure supplement 3A*). Conversely, in populations with a wider range of mean glucose levels, the HBGI correlates more strongly with mean glucose levels (*Figure 3A*). Despite these differences, our validation analyses confirm that CGM-derived indices consistently cluster into three components: mean, variance, and autocorrelation.

We acknowledge the potential concern of multiple testing in our study. However, even after adjusting for multiple comparisons, the CGM-derived indices retained significant correlations with %NC (*Figure 1E*). The consistency of our findings across different analytical approaches (Lasso, PLS, and factor analysis) and different data sets further supports the robustness of our conclusions regarding the characteristics of glucose dynamics. While we used methods that assume linearity, such as LASSO, we also examined nonlinear relationships using Spearman's correlation for index relationships and factor loadings with %NC and found significant associations. We acknowledge that some significant correlations appear to be relatively small. However, these findings, combined with our predictive models showing improved accuracy using CGM compared to traditional diabetes diagnostic indices, and the theoretical framework showing that conventional markers only consider the 'mean' component of glucose dynamics, can demonstrate the clinical significance. Although our analysis included four datasets with a total of 270 individuals, and our sample size of 53 met the required threshold based on power calculations with a type I error of 0.05, a power of 0.8, and an expected correlation coefficient of 0.4, we acknowledge that the sample size may still be considered relatively small for a comprehensive assessment of these relationships. To further validate these findings, larger prospective studies with diverse populations are needed. While CGM-derived indices, such as AC_Var and ADRR hold promise for CAD risk assessment, their complexity may present challenges for routine clinical implementation. To improve usability, we have developed a web-based calculator that automates these calculations. However, defining clinically relevant thresholds and reference ranges requires further validation in larger cohorts.

In conclusion, glucose dynamics have three components: mean, variance, and autocorrelation. These three components are associated with coronary plaque vulnerability. CGM-derived indices reflecting these three components can be valuable predictive tools for T2DM complications, compared to conventional diabetes diagnostic markers reflecting only the mean component. This new predictive model has the potential to improve the diagnosis and management of diabetes worldwide. To facilitate this CGM-derived prediction, we created a web application that performs a multiple regression model with these three components as input variables (https://cgmregressionapp2.streamlit.app/).

# Methods

**Key resources table**

| Reagent type (species) or resource | Designation | Source or reference | Identifiers | Additional information |
|---|---|---|---|---|
| Software, algorithm | SciPy v1.10.1 | *Virtanen et al., 2020* | RRID:SCR_008058 | |
| Software, algorithm | scikit-learn v1.0.2 | https://scikit-learn.org/stable/ | RRID:SCR_002577 | |

## Study design and population

This retrospective observational study was approved by the ethics committee of Kobe University Graduate School of Medicine (UMIN000018326; Kobe, Japan), as described previously (*Otowa-Suematsu et al., 2018*). Informed consent was obtained from all participants. The study included 53 participants who underwent a 75 g oral glucose tolerance test (OGTT), continuous glucose monitoring (CGM) with the use of an iPro2 CGM system (Medtronic, Northridge, CA, USA), and percutaneous coronary intervention (PCI). During PCI, VH-IVUS was carried out to assess the plaque components. All individuals underwent CGM for at least three consecutive days within the seven-day period prior to the PCI procedure.

Among the 53 participants, eight, 16, and 29 individuals were categorized as having normal glucose tolerance (NGT), impaired glucose tolerance (IGT), and T2DM, respectively. Of note, with a type I error of 0.05, a power of 0.8, and an expected correlation coefficient of 0.4, a sample size of 47 was required to detect a significant difference from zero in the correlation coefficient. T2DM was defined as HbA1c $\geq$ 6.5%, fasting plasma glucose (FPG) $\geq$ 126 mg/dL, or 2 hr plasma glucose during a 75 g OGTT (PG120) $\geq$ 200 mg/dL. IGT was defined as HbA1c 6.0–6.4%, FPG 110–125 mg/dL, or PG120 140–199 mg/dL. NGT was defined as values below all prediabetes thresholds (HbA1c < 6.0%, FPG < 110 mg/dL and PG120 < 140 mg/dL).

Detailed participant characteristics have been reported in the previous study (*Otowa-Suematsu et al., 2018*). Briefly, participants aged 20–80 years with LDL-C levels <120 mg/dL under statin administration or <100 mg/dL under other treatments for dyslipidemia, including lifestyle intervention, were included in this study. Participants with acute coronary syndrome, unsuitable anatomy for VH-IVUS, poor imaging by VH-IVUS, hemodialysis, inflammatory disease, shock, low cardiac output, or concurrent malignant disease were excluded from this study.

To validate the findings on glucose dynamics, a previously reported data set from Japan (*Sugimoto et al., 2025*) was analyzed. Informed consent was obtained from all participants. The study was approved by the Ethics Committee of Kobe University Hospital (approval number 1834). Briefly, individuals aged $\geq$20 years with no previous diagnosis of diabetes were recruited from Kobe University Hospital between January 2016 and March 2018. Exclusion criteria included the use of medications that affect glucose metabolism (e.g. steroids, β-blockers), psychiatric disorders, pregnancy or lactation, and ineligibility as determined by treating physicians. Participants wore a CGM device (iPro; Medtronic, USA). The study included 52 individuals with NGT, nine with IGT, and three with T2DM.

Further validation of the glucose dynamics findings was performed using a previously reported CGM dataset (Dexcom G4 CGM System; Dexcom, Fort Lauderdale, FL, USA) obtained in the United States (*Hall et al., 2018*). Informed consent was obtained from all participants. The study was approved by the Stanford Institutional Review Board (IRB 37141). The study included 53 individuals aged 30–70 years with a BMI of 23–40 kg/m$^2$. Exclusion criteria included prior diagnosis of diabetes, major organ diseases, uncontrolled hypertension, malignancy, chronic inflammatory conditions, use of any medications known to alter blood glucose or insulin sensitivity, haematocrit <30, creatinine above the normal range, and ALT levels more than twice the upper limit of normal.

In addition, the present study analyzed a previously reported CGM (FreeStyle Libre H, Abbott Diabetes Care, Witney, UK) dataset from China (*Zhao et al., 2023*). Participants were recruited from the DiaDRIL registry at Shanghai East Hospital (September 2019 to March 2021) and Shanghai Fourth People's Hospital (June 2021 to November 2021). Inclusion criteria for this dataset were: diagnosis of diabetes according to the 1999 World Health Organization (WHO) criteria, age 18 years or older, willingness to provide informed consent, and CGM recording for at least three days. Exclusion criteria included reported alcohol or drug abuse, inability to comply with study protocols, or inability to participate as determined by the investigators. We extracted and analyzed glucose profile characteristics from the first three days of CGM data for each participant. 100 individuals with

T2DM were analyzed in this study. Our primary objective was to explore the relationship between the CGM-derived indices and the presence of diabetic macrovascular complications. Of note, as the original reports document, the validation datasets did not specify explicit cutoffs for blood pressure or cholesterol. Consequently, they included participants with suboptimal control of these parameters.

The data collection protocol for the Chinese dataset was previously documented (*Zhao et al., 2023*). Briefly, trained research staff used standardized questionnaires to collect demographic and clinical information, including diabetes diagnosis, treatment history, comorbidities, and medication use. CGM records interstitial glucose levels at 15 min intervals for up to 14 days. Laboratory measurements, including metabolic panels, lipid profiles, and renal function tests, were obtained within six months of CGM placement. While previous studies have linked necrotic cores to macrovascular events, we acknowledge the limitations of the cardiovascular outcomes in the Chinese data set. These outcomes were extracted from medical records rather than from standardized diagnostic procedures or imaging studies.

## Calculation of clinical indices
### CGM-derived indices
The CGM data collected in this study were obtained at different sampling frequencies in the different data sets. For the two Japanese datasets and the American dataset, glucose measurements were collected at 5 min intervals. In contrast, the Chinese dataset collected glucose measurements at 15 min intervals.

Fourteen CGM-derived indices were evaluated in this study: twelve well-established CGM-derived indices (*Hill et al., 2011*) and two indices (AC_Mean and AC_Var) that have been reported to capture glucose handling capacity independently of the aforementioned twelve indices (*Sugimoto et al., 2025*). For the datasets with 5 min sampling intervals (the two Japanese datasets and the American dataset), AC_Mean and AC_Var were calculated as the mean and variance of the autocorrelation functions at lags 1–30 of the glucose levels, respectively. Of note, autocorrelation refers to the relationship between a variable and its past values over time. In the context of glucose dynamics, it reflects how current glucose levels are influenced by past levels, capturing patterns, such as sustained hyperglycemia or recurrent fluctuations. For example, if an individual experiences sustained high glucose levels after a meal, the strong correlation between successive glucose readings indicates high autocorrelation. For the data set with 15 min sampling intervals (the Chinese data set), AC_Mean and AC_Var were calculated as the mean and variance of the autocorrelation functions at lags 1–10 of the glucose levels, respectively. These calculations correspond to a time window of 150 min. CGM_Mean and CGM_Std indicate the mean value and standard deviation of glucose levels measured by CGM, respectively. CONGA, LI, JINDEX, HBGI, GRADE, MODD, MAGE, ADRR, MVALUE, and MAG were calculated using EasyGV software (*Hill et al., 2011*). The calculating formulae of these indices are shown in *Supplementary file 1*.

In the two Japanese datasets and the American dataset, there was a relatively small proportion of T2DM patients and a substantial number of individuals with time in range (TIR) (*Battelino et al., 2023*; *Larkin et al., 2019*) values (glucose levels between 70 and 180 mg/dL) greater than 70% (*Hall et al., 2018*; *Otowa-Suematsu et al., 2018*; *Sugimoto et al., 2025*). Given this distribution, we decided not to include TIR in our primary analyses. However, given the potential interest of this metric, we provided correlations between TIR and other indices as supplementary information.

### OGTT-derived indices
Three OGTT-derived indices were calculated as previously described (*Matsuda and DeFronzo, 1999*; *Otowa-Suematsu et al., 2018*). The insulinogenic index (I.I.) indicates insulin secretion and is calculated from the ratio of the increment of immunoreactive insulin (IRI) to that of plasma glucose at 30 min after onset of the OGTT. The composite index indicates insulin sensitivity, which can be calculated from fasting plasma glucose, fasting IRI, mean blood glucose levels, and mean serum IRI concentrations during the OGTT. The oral disposition index (Oral DI) was calculated from the product of the composite index and the ratio of the area under the insulin concentration curve from 0 to 120 min to that for plasma glucose from 0 to 120 min, without using the data measured at 90 min, in the OGTT.

## VH-IVUS-derived index

VH-IVUS was carried out using the Eagle Eye Platinum 3.5-Fr 20-MHz catheter (Volcano, Rancho Cordova, CA, USA), as previously described (*Otowa-Suematsu et al., 2018*). The intraclass correlation coefficients for interobserver and intraobserver reliability of external elastic membrane volume were 0.95 and 0.97, respectively (*Otowa-Suematsu et al., 2018*), indicating high reproducibility. The VH-IVUS categorized plaque into four components: fibrous, fibrofatty, necrotic core, and dense calcium. Following the previous study, our investigation focused specifically on the ratio of necrotic core to total plaque volume (%NC), a widely used parameter of plaque vulnerability. For patients with multiple plaques, the mean %NC was calculated.

## Prediction models and statistical analyses

We modelled %NC using multiple linear regression, LASSO regression, and PLS regression. The input variables in these models included the following 26 variables: BMI, SBP, DBP, TGs, LDL-C, HDL-C, FBG, HbA1c, PG120, I.I., composite index, oral DI, CGM_Mean, CGM_Std, CONGA, LI, JINDEX, HBGI, GRADE, MODD, MAGE, ADRR, MVALUE, MAG, AC_Mean, and AC_Var. In conducting these models, z-score normalization on each input variable was performed.

The predictive performance of multiple linear regression was evaluated by the coefficient of determination ($R^2$), the adjusted coefficient of determination (Adj $R^2$), or AIC. The multicollinearity of the input variables was estimated by VIF. LASSO regression is a kind of linear regression with L1 regularization (*Tibshirani, 1996*). The optimal regularization coefficient, lambda, was based on leave-one-out cross-validation and mean-squared error (MSE). The importance of the input variables in predicting %NC was evaluated by the VIP scores (*Wold et al., 2001*) that were generated from PLS regression. LASSO regression was chosen for its ability to perform feature selection by identifying the most relevant predictors. Unlike Ridge regression, which simply shrinks coefficients toward zero without reaching exactly zero, LASSO produces sparse models, which is consistent with our goal of identifying the most critical features of glucose dynamics associated with coronary plaque vulnerability. In addition, we implemented PLS regression as a complementary approach due to its effectiveness in dealing with multicollinearity, which was particularly relevant given the high correlation among several CGM-derived measures. These models were conducted using scikit-learn v1.0.2, a Python-based toolkit (https://scikit-learn.org/stable/).

Relationships among indices were also evaluated using Spearman's correlation coefficients (*r*), and the correlation coefficients were reported with 95% CIs through bootstrap resampling. The number of resamples performed to form the distribution was set at 10000. Benjamini–Hochberg's multiple comparison test was also performed with a significance threshold of Q<0.05.

## Factor analysis and hierarchical clustering analysis

The intercorrelations of the clinical parameters and their associations with %NC were assessed using exploratory factor analyses and hierarchical clustering analyses. We followed the previously reported approach (*Cappelleri et al., 2000*; *Guo et al., 2020*; *Lakka et al., 2002*; *Oh et al., 2004*) with some modifications in conducting our exploratory factor analyses. BIC and MAP methods were used to determine the number of underlying factors. Variables with absolute factor loadings of ≥0.30 were used in interpretation. To improve the interpretation, orthogonal (varimax) rotation was used. To evaluate the applicability of the factor analysis, KMO and Bartlett's spherical test were performed. To evaluate the internal consistency of each factor, Cronbach's α and item–total correlations were calculated. The association of the factor scores with %NC was assessed using Spearman's correlation.

Hierarchical clustering analysis was conducted using a method that combines a Euclidean distance measure and Ward linkage. I.I., composite index, oral DI, and AC_Mean were inverted negatively so that the value of indices increased in individuals with abnormalities. The quality of the hierarchical clustering analysis was evaluated based on silhouette analysis (*Rousseeuw, 1987*). These analyses were performed after Z-score normalization using scikit-learn v1.0.2, a Python-based toolkit (https://scikit-learn.org/stable/).

## Mathematical models used for simulating the characteristics of glucose dynamics

In simulating the characteristics of glucose dynamics, we used a simple and stable model (*De Gaetano and Arino, 2000*), which can be written as follows:

$$\frac{dG}{dt} = -k_{\mathrm{glu}}G - k_{\mathrm{sen}}IG + k_{\mathrm{pro}} + f$$

$$\frac{dI}{dt} = \frac{k_{\mathrm{sec}}}{k_{\mathrm{tim}}} \int_{t-k_{\mathrm{tim}}}^{t} G\,ds - k_{\mathrm{cle}}I$$

where the variables $G$ and $I$ denote blood glucose and insulin concentrations, respectively. We simulated 240 min profiles of $G$, and calculated the mean, Std, and AC_Var of $G$. The parameters were changed within the range participants could take (*De Gaetano and Arino, 2000*). Five mg/dL/min glucose was applied for 10 min at 30 min as the external input of glucose $f$. The simulations were conducted using SciPy v1.10.1 (*Virtanen et al., 2020*).

## Characterization of glucose patterns during the OGTT

The relation between the three components of glucose dynamics and the characteristics of previously reported glucose patterns during the OGTT (*Hulman et al., 2018*) was investigated. In the study, 5861 individuals without diabetes in Denmark underwent the OGTT with measurements of glucose levels at three time points (0, 30, and 120 min), and four distinct glucose patterns associated with long-term outcomes including diabetes onset, cardiovascular disease, and all-cause mortality rate were identified. For the calculation of mean, Std, and AC_Var of glucose levels, each time point was linearly interpolated. AC_Var was calculated using the autocorrelation function at lags 1–20 because glucose data were available only for 2 hr after the OGTT.

## Acknowledgements

This study was supported by the Japan Society for the Promotion of Science (JSPS) KAKENHI (JP21H04759), CREST, the Japan Science and Technology Agency (JST) (JPMJCR2123), The Uehara Memorial Foundation, and The Takeda Science Foundation.

## Additional information

### Funding

| Funder | Grant reference number | Author |
| --- | --- | --- |
| Japan Society for the Promotion of Science | JP21H04759 | Shinya Kuroda |
| Japan Science and Technology Agency | 10.52926/jpmjcr2123 | Shinya Kuroda |
| Uehara Memorial Foundation | | Shinya Kuroda |
| Takeda Science Foundation | | Hikaru Sugimoto |

The funders had no role in study design, data collection and interpretation, or the decision to submit the work for publication.

### Author contributions

Hikaru Sugimoto, Conceptualization, Software, Formal analysis, Investigation, Visualization, Methodology, Writing – original draft; Ken-ichi Hironaka, Yushi Hirota, Hiromasa Otake, Ken-Ichi Hirata, Kazuhiko Sakaguchi, Writing – review and editing; Tomoko Yamada, Natsu Otowa-Suematsu, Data curation, Writing – review and editing; Wataru Ogawa, Shinya Kuroda, Project administration, Writing – review and editing

## Author ORCIDs
Hikaru Sugimoto (iD) https://orcid.org/0000-0002-6468-0127
Shinya Kuroda (iD) https://orcid.org/0000-0001-5059-8299

## Ethics
Informed consent was obtained from all participants. This study was approved by the ethics committee of Kobe University Graduate School of Medicine (UMIN000018326; Kobe, Japan), the Ethics Committee of Kobe University Hospital (approval number 1834), and the Stanford Institutional Review Board (IRB 37141).

Reviewer #2 (Public review): https://doi.org/10.7554/eLife.102860.3.sa1
Reviewer #3 (Public review): https://doi.org/10.7554/eLife.102860.3.sa2
Author response https://doi.org/10.7554/eLife.102860.3.sa3

# Additional files

## Supplementary files
MDAR checklist

Supplementary file 1. Calculating formulae of the continuous glucose monitoring-derived indices.

## Data availability
We used only previously published data sets. The code that calculates AC_Mean and AC_Var and that performs regression analysis with CGM-derived indices as input variables are available from the repository (https://github.com/HikaruSugimoto/CGM_regression_app, copy archived at *Sugimoto, 2026*) and the web application (https://cgmregressionapp2.streamlit.app/).

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
