## [Editor Report · eLife Assessment]

This **valuable** retrospective analysis identified three independent components of glucose dynamics - "value", "variability", and "autocorrelation" - which may be used in predicting coronary plaque vulnerability. The study is **solid** and of interest to a wide range of investigators in the medical field who are interested in the role of glycemia on cardiometabolic health. The manuscript has been substantially strengthened by clarifying methods, improving transparency, and validating key findings, resulting in a coherent and persuasive case for autocorrelation as a meaningful third dimension of glucose dynamics despite remaining design-related limitations.

---

## [Referee Report · Reviewer #2 (Public review)]

Summary:

Sugimoto et al. explore the relationship between glucose dynamics-specifically value, variability, and autocorrelation-and coronary plaque vulnerability in patients with varying glucose tolerance levels. The study identifies three independent predictive factors for %NC and emphasizes the use of continuous glucose monitoring (CGM)-derived indices for coronary artery disease (CAD) risk assessment. By employing robust statistical methods and validating findings across datasets from Japan, America, and China, the authors highlight the limitations of conventional markers while proposing CGM as a novel approach for risk prediction.The study has the potential to reshape CAD risk assessment by emphasizing CGM-derived indices, aligning well with personalized medicine trends.

Further, the revised version includes expanded biological interpretation, improved statistical justification, and a new web-based calculator for clinical translation. Together, these updates make the study an important contribution to precision risk assessment in diabetes and cardiovascular research.

Strengths:

The introduction of autocorrelation as a predictive factor for plaque vulnerability adds a novel dimension to glucose dynamic analysis.

Inclusion of datasets from diverse regions enhances generalizability.

The use of a well-characterized cohort with controlled cholesterol and blood pressure levels strengthens the findings.

The focus on CGM-derived indices aligns with personalized medicine trends, showcasing potential for CAD risk stratification.

The benchmarking of CGM-derived measures against established CAD risk models (e.g., Framingham Risk Score) enhances interpretability and significance.

The addition of a web-based computational tool makes the proposed indices accessible for potential clinical and research use.

Weaknesses:

The biological mechanism linking glucose autocorrelation to plaque vulnerability, although plausibly associated with insulin clearance pathways, remains largely theoretical.

The primary cohort size is still modest, and while supported by power analysis and external datasets, broader prospective validation will be important.

Strict participant selection criteria as employed by the study may reduce applicability to broader populations.

CGM-derived indices like AC_Var and ADRR may be too complex for routine clinical use without simplified models or guidelines.

Comments on revised version:

The authors have thoroughly addressed previous concerns and produced a much stronger manuscript. The study now provides a coherent, validated, and well-reasoned argument for including autocorrelation as a third major dimension of glucose dynamics. It offers both conceptual novelty and translational potential and will likely stimulate further research on temporal glucose metrics in metabolic and cardiovascular risk assessment.

---

## [Referee Report · Reviewer #3 (Public review)]

Summary:

This is a retrospective analysis of 53 individuals over 26 features (12 clinical phenotypes, 12 CGM features, and 2 autocorrelation features) to examine which features were most informative in predicting percent necrotic core (%NC) as parameter for coronary plaque vulnerability. Multiple regression analysis demonstrated a better ability to predict %NC from 3 selected CGM derived features than 3 selected clinical phenotypes. LASSO regularization and partial least squares (PLS) with VIP scores were used to identify 4 CGM features that most contribute to the precision of %NC. Using factor analysis they identify 3 components that have CGM related features: value (relating to the value of blood glucose), variability (relating to glucose variability), and autocorrelation (composed of the two autocorrelation features). These three groupings appeared in the 3 validation cohorts and when performing hierarchical clustering. To demonstrate how these three features change, a simulation was created to allow the user to examine these features under different conditions.

Summary of Revision 1. This is a Valuable study supported by Solid evidence. The revisions meaningfully strengthen the manuscript by clarifying methods, improving transparency, and refining presentation. The work provides useful conceptual and methodological advances for understanding CGM-derived glucose dynamics and their possible relationship to cardiovascular pathology.

Strengths:

The authors have provided a much clearer exposition of how each glycemic component was defined and validated across cohorts. The revised manuscript now includes explicit pairwise correlations, clarified p- and q-value reporting, and better visualization of key associations between CGM indices and %NC. The justification for LASSO and PLS use is now well explained, and additional details on cohort timing relative to PCI, validation dataset structure, and statistical robustness (e.g., VIP stability with covariates) address prior concerns. The inclusion of precise factor definitions and clearer graphics notably improves interpretability.

Limitations:

Some limitations remain inherent to the study design, including the modest primary sample size, reliance on retrospective data, and differences between validation datasets in outcome ascertainment. However, these are now acknowledged more openly.

---

## [Author Response]

The following is the authors’ response to the original reviews.

**Public Reviews:**

**Reviewer #1 (Public review):**

We appreciate the reviewer for the critical review of the manuscript and the valuable comments. We have carefully considered the reviewer’s comments and have revised our manuscript accordingly.

The reviewer’s comments in this letter are in Bold and Italics.

Summary:This study identified three independent components of glucose dynamics-"value," "variability," and "autocorrelation", and reported important findings indicating that they play an important role in predicting coronary plaque vulnerability. Although the generalizability of the results needs further investigation due to the limited sample size and validation cohort limitations, this study makes several notable contributions: validation of autocorrelation as a new clinical indicator, theoretical support through mathematical modeling, and development of a web application for practical implementation. These contributions are likely to attract broad interest from researchers in both diabetology and cardiology and may suggest the potential for a new approach to glucose monitoring that goes beyond conventional glycemic control indicators in clinical practice.Strengths:The most notable strength of this study is the identification of three independent elements in glycemic dynamics: value, variability, and autocorrelation. In particular, the metric of autocorrelation, which has not been captured by conventional glycemic control indices, may bring a new perspective for understanding glycemic dynamics. In terms of methodological aspects, the study uses an analytical approach combining various statistical methods such as factor analysis, LASSO, and PLS regression, and enhances the reliability of results through theoretical validation using mathematical models and validation in other cohorts. In addition, the practical aspect of the research results, such as the development of a Web application, is also an important contribution to clinical implementation.

We appreciate reviewer #1 for the positive assessment and for the valuable and constructive comments on our manuscript.

Weaknesses:The most significant weakness of this study is the relatively small sample size of 53 study subjects. This sample size limitation leads to a lack of statistical power, especially in subgroup analyses, and to limitations in the assessment of rare events.

We appreciate the reviewer’s concern regarding the sample size. We acknowledge that a larger sample size would increase statistical power, especially for subgroup analyses and the assessment of rare events.

We would like to clarify several points regarding the statistical power and validation of our findings. Our sample size determination followed established methodological frameworks, including the guidelines outlined by Muyembe Asenahabi, Bostely, and Peters Anselemo Ikoha. “Scientific research sample size determination.” (2023). These guidelines balance the risks of inadequate sample size with the challenges of unnecessarily large samples. For our primary analysis examining the correlation between CGM-derived measures and %NC, power calculations (a type I error of 0.05, a power of 0.8, and an expected correlation coefficient of 0.4) indicated that a minimum of 47 participants was required. Our sample size of 53 exceeded this threshold and allowed us to detect statistically significant correlations, as described in the Methods section. Moreover, to provide transparency about the precision of our estimates, we have included confidence intervals for all coefficients.

Furthermore, our sample size aligns with previous studies investigating the associations between glucose profiles and clinical parameters, including Torimoto, Keiichi, et al. “Relationship between fluctuations in glucose levels measured by continuous glucose monitoring and vascular endothelial dysfunction in type 2 diabetes mellitus.” Cardiovascular Diabetology 12 (2013): 1-7. (n=57), Hall, Heather, et al. “Glucotypes reveal new patterns of glucose dysregulation.” PLoS biology 16.7 (2018): e2005143. (n=57), and Metwally, Ahmed A., et al. “Prediction of metabolic subphenotypes of type 2 diabetes via continuous glucose monitoring and machine learning.” Nature Biomedical Engineering (2024): 1-18. (n=32).

Furthermore, the primary objective of our study was not to assess rare events, but rather to demonstrate that glucose dynamics can be decomposed into three main factors - mean, variance and autocorrelation - whereas traditional measures have primarily captured mean and variance without adequately reflecting autocorrelation. We believe that our current sample size effectively addresses this objective.

Regarding the classification of glucose dynamics components, we have conducted additional validation across diverse populations including 64 Japanese, 53 American, and 100 Chinese individuals. These validation efforts have consistently supported our identification of three independent glucose dynamics components.

However, we acknowledge the importance of further validation on a larger scale. To address this, we conducted a large follow-up study of over 8,000 individuals (Sugimoto, Hikaru, et al. “Stratification of individuals without prior diagnosis of diabetes using continuous glucose monitoring” medRxiv (2025)), which confirmed our main finding that glucose dynamics consist of mean, variance, and autocorrelation. As this large study was beyond the scope of the present manuscript due to differences in primary objectives and analytical approaches, it was not included in this paper; however, it provides further support for the clinical relevance and generalizability of our findings.

To address the sample size considerations, we have added the following sentences in the Discussion section (lines 409-414):

Although our analysis included four datasets with a total of 270 individuals, and our sample size of 53 met the required threshold based on power calculations with a type I error of 0.05, a power of 0.8, and an expected correlation coefficient of 0.4, we acknowledge that the sample size may still be considered relatively small for a comprehensive assessment of these relationships. To further validate these findings, larger prospective studies with diverse populations are needed.

We appreciate the reviewer’s feedback and believe that these clarifications improve the manuscript.

In terms of validation, several challenges exist, including geographical and ethnic biases in the validation cohorts, lack of long-term follow-up data, and insufficient validation across different clinical settings. In terms of data representativeness, limiting factors include the inclusion of only subjects with well-controlled serum cholesterol and blood pressure and the use of only short-term measurement data.

We appreciate the reviewer’s comment regarding the challenges associated with validation. In terms of geographic and ethnic diversity, our study includes validation datasets from diverse populations, including 64 Japanese, 53 American and 100 Chinese individuals. These datasets include a wide range of metabolic states, from healthy individuals to those with diabetes, ensuring validation across different clinical conditions. In addition, we recognize the limited availability of publicly available datasets with sufficient sample sizes for factor decomposition that include both healthy individuals and those with type 2 diabetes (Zhao, Qinpei, et al. “Chinese diabetes datasets for data-driven machine learning.” Scientific Data 10.1 (2023): 35.). The main publicly available datasets with relevant clinical characteristics have already been analyzed in this study using unbiased approaches.

However, we fully agree with the reviewer that expanding the geographic and ethnic scope, including long-term follow-up data, and validation in different clinical settings would further strengthen the robustness and generalizability of our findings. To address this, we conducted a large follow-up study of over 8,000 individuals with two years of follow-up (Sugimoto, Hikaru, et al. “Stratification of individuals without prior diagnosis of diabetes using continuous glucose monitoring” medRxiv (2025)), which confirmed our main finding that glucose dynamics consist of mean, variance, and autocorrelation. As this large study was beyond the scope of the present manuscript due to differences in primary objectives and analytical approaches, it was not included in this paper; however, it provides further support for the clinical relevance and generalizability of our findings.

Regarding the validation considerations, we have added the following sentences to the Discussion section (lines 409-414, 354-361):

Although our analysis included four datasets with a total of 270 individuals, and our sample size of 53 met the required threshold based on power calculations with a type I error of 0.05, a power of 0.8, and an expected correlation coefficient of 0.4, we acknowledge that the sample size may still be considered relatively small for a comprehensive assessment of these relationships. To further validate these findings, larger prospective studies with diverse populations are needed.

Although our LASSO and factor analysis indicated that CGM-derived measures were strong predictors of %NC, this does not mean that other clinical parameters, such as lipids and blood pressure, are irrelevant in T2DM complications. Our study specifically focused on characterizing glucose dynamics, and we analyzed individuals with well-controlled serum cholesterol and blood pressure to reduce confounding effects. While we anticipate that inclusion of a more diverse population would not alter our primary findings regarding glucose dynamics, it is likely that a broader data set would reveal additional predictive contributions from lipid and blood pressure parameters.

In terms of elucidation of physical mechanisms, the study is not sufficient to elucidate the mechanisms linking autocorrelation and clinical outcomes or to verify them at the cellular or molecular level.

We appreciate the reviewer’s point regarding the need for further elucidation of the physical mechanisms linking glucose autocorrelation to clinical outcomes. We fully agree with the reviewer that the detailed molecular and cellular mechanisms underlying this relationship are not yet fully understood, as noted in our Discussion section.

However, we would like to emphasize the theoretical basis that supports the clinical relevance of autocorrelation. Our results show that glucose profiles with identical mean and variability can exhibit different autocorrelation patterns, highlighting that conventional measures such as mean or variance alone may not fully capture inter-individual metabolic differences. Incorporating autocorrelation analysis provides a more comprehensive characterization of metabolic states. Consequently, incorporating autocorrelation measures alongside traditional diabetes diagnostic criteria - such as fasting glucose, HbA1c and PG120, which primarily reflect only the “mean” component - can improve predictive accuracy for various clinical outcomes. While further research at the cellular and molecular level is needed to fully validate these findings, it is important to note that the primary goal of this study was to analyze the characteristics of glucose dynamics and gain new insights into metabolism, rather than to perform molecular biology experiments.

Furthermore, our previous research has shown that glucose autocorrelation reflects changes in insulin clearance (Sugimoto, Hikaru, et al. “Improved detection of decreased glucose handling capacities via continuous glucose monitoring-derived indices.” Communications Medicine 5.1 (2025): 103.). The relationship between insulin clearance and cardiovascular disease has been well documented (Randrianarisoa, Elko, et al. “Reduced insulin clearance is linked to subclinical atherosclerosis in individuals at risk for type 2 diabetes mellitus.” Scientific reports 10.1 (2020): 22453.), and the mechanisms described in this prior work may potentially explain the association between glucose autocorrelation and clinical outcomes observed in the present study.

Rather than a limitation, we view these currently unexplored associations as an opportunity for further research. The identification of autocorrelation as a key glycemic feature introduces a new dimension to metabolic regulation that could serve as the basis for future investigations exploring the molecular mechanisms underlying these patterns.

While we agree that further research at the cellular and molecular level is needed to fully validate these findings, we believe that our study provides a theoretical framework to support the clinical utility of autocorrelation analysis in glucose monitoring, and that this could serve as the basis for future investigations exploring the molecular mechanisms underlying these autocorrelation patterns, which adds to the broad interest of this study. Regarding the physical mechanisms linking autocorrelation and clinical outcomes, we have added the following sentences in the Discussion section (lines 331-339, 341-352):

This study also provided evidence that autocorrelation can vary independently from the mean and variance components using simulated data. In addition, simulated glucose dynamics indicated that even individuals with high AC_Var did not necessarily have high maximum and minimum blood glucose levels. This study also indicated that these three components qualitatively corresponded to the four distinct glucose patterns observed after glucose administration, which were identified in a previous study (Hulman et al., 2018). Thus, the inclusion of autocorrelation in addition to mean and variance may improve the characterization of inter-individual differences in glucose regulation and improve the predictive accuracy of various clinical outcomes.

Despite increasing evidence linking glycemic variability to oxidative stress and endothelial dysfunction in T2DM complications (Ceriello et al., 2008; Monnier et al., 2008), the biological mechanisms underlying the independent predictive value of autocorrelation remain to be elucidated. Our previous work has shown that glucose autocorrelation is influenced by insulin clearance (Sugimoto et al., 2025), a process known to be associated with cardiovascular disease risk (Randrianarisoa et al., 2020). Therefore, the molecular pathways linking glucose autocorrelation to cardiovascular disease may share common mechanisms with those linking insulin clearance to cardiovascular disease. Although previous studies have primarily focused on investigating the molecular mechanisms associated with mean glucose levels and glycemic variability, our findings open new avenues for exploring the molecular basis of glucose autocorrelation, potentially revealing novel therapeutic targets for preventing diabetic complications.

**Reviewer #2 (Public review):**

We appreciate the reviewer for the critical review of the manuscript and the valuable comments. We have carefully considered the reviewer’s comments and have revised our manuscript accordingly. The reviewer’s comments in this letter are in Bold and Italics.

Sugimoto et al. explore the relationship between glucose dynamics - specifically value, variability, and autocorrelation - and coronary plaque vulnerability in patients with varying glucose tolerance levels. The study identifies three independent predictive factors for %NC and emphasizes the use of continuous glucose monitoring (CGM)-derived indices for coronary artery disease (CAD) risk assessment. By employing robust statistical methods and validating findings across datasets from Japan, America, and China, the authors highlight the limitations of conventional markers while proposing CGM as a novel approach for risk prediction. The study has the potential to reshape CAD risk assessment by emphasizing CGM-derived indices, aligning well with personalized medicine trends.Strengths:(1) The introduction of autocorrelation as a predictive factor for plaque vulnerability adds a novel dimension to glucose dynamic analysis.(2) Inclusion of datasets from diverse regions enhances generalizability.(3) The use of a well-characterized cohort with controlled cholesterol and blood pressure levels strengthens the findings.(4) The focus on CGM-derived indices aligns with personalized medicine trends, showcasing the potential for CAD risk stratification.

We appreciate reviewer #2 for the positive assessment and for the valuable and constructive comments on our manuscript.

Weaknesses:(1) The link between autocorrelation and plaque vulnerability remains speculative without a proposed biological explanation.

We appreciate the reviewer’s point about the need for a clearer biological explanation linking glucose autocorrelation to plaque vulnerability. We fully agree with the reviewer that the detailed biological mechanisms underlying this relationship are not yet fully understood, as noted in our Discussion section.

However, we would like to emphasize the theoretical basis that supports the clinical relevance of autocorrelation. Our results show that glucose profiles with identical mean and variability can exhibit different autocorrelation patterns, highlighting that conventional measures such as mean or variance alone may not fully capture inter-individual metabolic differences. Incorporating autocorrelation analysis provides a more comprehensive characterization of metabolic states. Consequently, incorporating autocorrelation measures alongside traditional diabetes diagnostic criteria - such as fasting glucose, HbA1c and PG120, which primarily reflect only the “mean” component - can improve predictive accuracy for various clinical outcomes.

Furthermore, our previous research has shown that glucose autocorrelation reflects changes in insulin clearance (Sugimoto, Hikaru, et al. “Improved detection of decreased glucose handling capacities via continuous glucose monitoring-derived indices.” Communications Medicine 5.1 (2025): 103.). The relationship between insulin clearance and cardiovascular disease has been well documented (Randrianarisoa, Elko, et al. “Reduced insulin clearance is linked to subclinical atherosclerosis in individuals at risk for type 2 diabetes mellitus.” Scientific reports 10.1 (2020): 22453.), and the mechanisms described in this prior work may potentially explain the association between glucose autocorrelation and clinical outcomes observed in the present study.

Rather than a limitation, we view these currently unexplored associations as an opportunity for further research. The identification of autocorrelation as a key glycemic feature introduces a new dimension to metabolic regulation that could serve as the basis for future investigations exploring the molecular mechanisms underlying these patterns.

While we agree that further research at the cellular and molecular level is needed to fully validate these findings, we believe that our study provides a theoretical framework to support the clinical utility of autocorrelation analysis in glucose monitoring, and that this could serve as the basis for future investigations exploring the molecular mechanisms underlying these autocorrelation patterns, which adds to the broad interest of this study. Regarding the physical mechanisms linking autocorrelation and clinical outcomes, we have added the following sentences in the Discussion section (lines 331-339, 341-352):

This study also provided evidence that autocorrelation can vary independently from the mean and variance components using simulated data. In addition, simulated glucose dynamics indicated that even individuals with high AC_Var did not necessarily have high maximum and minimum blood glucose levels. This study also indicated that these three components qualitatively corresponded to the four distinct glucose patterns observed after glucose administration, which were identified in a previous study (Hulman et al., 2018). Thus, the inclusion of autocorrelation in addition to mean and variance may improve the characterization of inter-individual differences in glucose regulation and improve the predictive accuracy of various clinical outcomes.

Despite increasing evidence linking glycemic variability to oxidative stress and endothelial dysfunction in T2DM complications (Ceriello et al., 2008; Monnier et al., 2008), the biological mechanisms underlying the independent predictive value of autocorrelation remain to be elucidated. Our previous work has shown that glucose autocorrelation is influenced by insulin clearance (Sugimoto et al., 2025), a process known to be associated with cardiovascular disease risk (Randrianarisoa et al., 2020). Therefore, the molecular pathways linking glucose autocorrelation to cardiovascular disease may share common mechanisms with those linking insulin clearance to cardiovascular disease. Although previous studies have primarily focused on investigating the molecular mechanisms associated with mean glucose levels and glycemic variability, our findings open new avenues for exploring the molecular basis of glucose autocorrelation, potentially revealing novel therapeutic targets for preventing diabetic complications.

(2) The relatively small sample size (n=270) limits statistical power, especially when stratified by glucose tolerance levels.

We appreciate the reviewer’s concern regarding sample size and its potential impact on statistical power, especially when stratified by glucose tolerance levels. We fully agree that a larger sample size would increase statistical power, especially for subgroup analyses.

We would like to clarify several points regarding the statistical power and validation of our findings. Our sample size followed established methodological frameworks, including the guidelines outlined by Muyembe Asenahabi, Bostely, and Peters Anselemo Ikoha. “Scientific research sample size determination.” (2023). These guidelines balance the risks of inadequate sample size with the challenges of unnecessarily large samples. For our primary analysis examining the correlation between CGM-derived measures and %NC, power calculations (a type I error of 0.05, a power of 0.8, and an expected correlation coefficient of 0.4) indicated that a minimum of 47 participants was required. Our sample size of 53 exceeded this threshold and allowed us to detect statistically significant correlations, as described in the Methods section. Moreover, to provide transparency about the precision of our estimates, we have included confidence intervals for all coefficients.

Furthermore, our sample size aligns with previous studies investigating the associations between glucose profiles and clinical parameters, including Torimoto, Keiichi, et al. “Relationship between fluctuations in glucose levels measured by continuous glucose monitoring and vascular endothelial dysfunction in type 2 diabetes mellitus.” Cardiovascular Diabetology 12 (2013): 1-7. (n=57), Hall, Heather, et al. “Glucotypes reveal new patterns of glucose dysregulation.” PLoS biology 16.7 (2018): e2005143. (n=57), and Metwally, Ahmed A., et al. “Prediction of metabolic subphenotypes of type 2 diabetes via continuous glucose monitoring and machine learning.” Nature Biomedical Engineering (2024): 1-18. (n=32).

Regarding the classification of glucose dynamics components, we have conducted additional validation across diverse populations including 64 Japanese, 53 American, and 100 Chinese individuals. These validation efforts have consistently supported our identification of three independent glucose dynamics components.

However, we acknowledge the importance of further validation on a larger scale. To address this, we conducted a large follow-up study of over 8,000 individuals with two years of followup (Sugimoto, Hikaru, et al. “Stratification of individuals without prior diagnosis of diabetes using continuous glucose monitoring” medRxiv (2025)), which confirmed our main finding that glucose dynamics consist of mean, variance, and autocorrelation. As this large study was beyond the scope of the present manuscript due to differences in primary objectives and analytical approaches, it was not included in this paper; however, it provides further support for the clinical relevance and generalizability of our findings.

To address the sample size considerations, we have added the following sentences in the Discussion section (lines 409-414):

Although our analysis included four datasets with a total of 270 individuals, and our sample size of 53 met the required threshold based on power calculations with a type I error of 0.05, a power of 0.8, and an expected correlation coefficient of 0.4, we acknowledge that the sample size may still be considered relatively small for a comprehensive assessment of these relationships. To further validate these findings, larger prospective studies with diverse populations are needed.

(3) Strict participant selection criteria may reduce applicability to broader populations**.**

We appreciate the reviewer’s comment regarding the potential impact of strict participant selection criteria on the broader applicability of our findings. We acknowledge that extending validation to more diverse populations would improve the generalizability of our findings.

Our study includes validation cohorts from diverse populations, including 64 Japanese, 53 American and 100 Chinese individuals. These cohorts include a wide range of metabolic states, from healthy individuals to those with diabetes, ensuring validation across different clinical conditions. However, we acknowledge that further validation in additional populations and clinical settings would strengthen our conclusions. To address this, we conducted a large follow-up study of over 8,000 individuals (Sugimoto, Hikaru, et al. “Stratification of individuals without prior diagnosis of diabetes using continuous glucose monitoring” medRxiv (2025)), which confirmed our main finding that glucose dynamics consist of mean, variance, and autocorrelation. As this large study was beyond the scope of the present manuscript due to differences in primary objectives and analytical approaches, it was not included in this paper; however, it provides further support for the clinical relevance and generalizability of our findings.

We have added the following text to the Discussion section to address these considerations (lines 409-414, 354-361):

Although our analysis included four datasets with a total of 270 individuals, and our sample size of 53 met the required threshold based on power calculations with a type I error of 0.05, a power of 0.8, and an expected correlation coefficient of 0.4, we acknowledge that the sample size may still be considered relatively small for a comprehensive assessment of these relationships. To further validate these findings, larger prospective studies with diverse populations are needed.

Although our LASSO and factor analysis indicated that CGM-derived measures were strong predictors of %NC, this does not mean that other clinical parameters, such as lipids and blood pressure, are irrelevant in T2DM complications. Our study specifically focused on characterizing glucose dynamics, and we analyzed individuals with well-controlled serum cholesterol and blood pressure to reduce confounding effects. While we anticipate that inclusion of a more diverse population would not alter our primary findings regarding glucose dynamics, it is likely that a broader data set would reveal additional predictive contributions from lipid and blood pressure parameters.

(4) CGM-derived indices like AC_Var and ADRR may be too complex for routine clinical use without simplified models or guidelines.

We appreciate the reviewer’s concern about the complexity of CGM-derived indices such as AC_Var and ADRR for routine clinical use. We acknowledge that for these indices to be of practical use, they must be both interpretable and easily accessible to healthcare providers.

To address this concern, we have developed an easy-to-use web application that automatically calculates these measures, including AC_Var, mean glucose levels, and glucose variability (https://cgmregressionapp2.streamlit.app/). This tool eliminates the need for manual calculations, making these indices more practical for clinical implementation.

Regarding interpretability, we acknowledge that establishing specific clinical guidelines would enhance the practical utility of these measures. For example, defining a cut-off value for AC_Var above which the risk of diabetes complications increases significantly would provide clearer clinical guidance. However, given our current sample size limitations and our predefined objective of investigating correlations among indices, we have taken a conservative approach by focusing on the correlation between AC_Var and %NC rather than establishing definitive cutoffs. This approach intentionally avoids problematic statistical practices like phacking. It is not realistic to expect a single study to accomplish everything from proposing a new concept to conducting large-scale clinical trials to establishing clinical guidelines. Establishing clinical guidelines typically requires the accumulation of multiple studies over many years. Recognizing this reality, we have been careful in our manuscript to make modest claims about the discovery of new “correlations” rather than exaggerated claims about immediate routine clinical use.

To address this limitation, we conducted a large follow-up study of over 8,000 individuals in the next study (Sugimoto, Hikaru, et al. “Stratification of individuals without prior diagnosis of diabetes using continuous glucose monitoring” medRxiv (2025)), which proposed clinically relevant cutoffs and reference ranges for AC_Var and other CGM-derived indices. As this large study was beyond the scope of the present manuscript due to differences in primary objectives and analytical approaches, it was not included in this paper; however, by integrating automated calculation tools with clear clinical thresholds, we expect to make these measures more accessible for clinical use.

We have added the following text to the Discussion section to address these considerations (lines 415-419):

While CGM-derived indices such as AC_Var and ADRR hold promise for CAD risk assessment, their complexity may present challenges for routine clinical implementation. To improve usability, we have developed a web-based calculator that automates these calculations. However, defining clinically relevant thresholds and reference ranges requires further validation in larger cohorts.

(5) The study does not compare CGM-derived indices to existing advanced CAD risk models, limiting the ability to assess their true predictive superiority.

We appreciate the reviewer’s comment regarding the comparison of CGMderived indices with existing CAD risk models. Given that our study population consisted of individuals with well-controlled total cholesterol and blood pressure levels, a direct comparison with the Framingham Risk Score for Hard Coronary Heart Disease (Wilson, Peter WF, et al. “Prediction of coronary heart disease using risk factor categories.” Circulation 97.18 (1998): 1837-1847.) may introduce inherent bias, as these factors are key components of the score.

Nevertheless, to further assess the predictive value of the CGM-derived indices, we performed additional analyses using linear regression to predict %NC. Using the Framingham Risk Score, we obtained an R² of 0.04 and an Akaike Information Criterion (AIC) of 330. In contrast, our proposed model incorporating the three glycemic parameters - CGM_Mean, CGM_Std, and AC_Var - achieved a significantly improved R² of 0.36 and a lower AIC of 321, indicating superior predictive accuracy.

We have added the following text to the Result section (lines 115-122):

The regression model including CGM_Mean, CGM_Std and AC_Var to predict %NC achieved an R² of 0.36 and an Akaike Information Criterion (AIC) of 321. Each of these indices showed statistically significant independent positive correlations with %NC (Fig. 1A). In contrast, the model using conventional glycemic markers (FBG, HbA1c, and PG120) yielded an R² of only 0.05 and an AIC of 340 (Fig. 1B). Similarly, the model using the Framingham Risk Score for Hard Coronary Heart Disease (Wilson et al., 1998) showed limited predictive value, with an R² of 0.04 and an AIC of 330 (Fig. 1C).

(6) Varying CGM sampling intervals (5-minute vs. 15-minute) were not thoroughly analyzed for impact on results**.**

We appreciate the reviewer’s comment regarding the potential impact of different CGM sampling intervals on our results. To assess the robustness of our findings across different sampling frequencies, we performed a down sampling analysis by converting our 5minute interval data to 15-minute intervals. The AC_Var value calculated from 15-minute intervals was significantly correlated with that calculated from 5-minute intervals (R = 0.99, 95% CI: 0.97-1.00). Furthermore, the regression model using CGM_Mean, CGM_Std, and AC_Var from 15-minute intervals to predict %NC achieved an R² of 0.36 and an AIC of 321, identical to the model using 5-minute intervals. These results indicate that our results are robust to variations in CGM sampling frequency.

We have added this analysis to the Result section (lines 122-125):

The AC_Var computed from 15-minute CGM sampling was nearly identical to that computed from 5-minute sampling (R = 0.99, 95% CI: 0.97-1.00) (Fig. S1A), and the regression using the 15‑min features yielded almost the same performance (R² = 0.36; AIC = 321; Fig. S1B).

**Reviewer #3 (Public review):**

We appreciate the reviewer for the critical review of the manuscript and the valuable comments. We have carefully considered the reviewer’s comments and have revised our manuscript accordingly. The reviewer’s comments in this letter are in Bold and Italics.

Summary:This is a retrospective analysis of 53 individuals over 26 features (12 clinical phenotypes, 12 CGM features, and 2 autocorrelation features) to examine which features were most informative in predicting percent necrotic core (%NC) as a parameter for coronary plaque vulnerability. Multiple regression analysis demonstrated a better ability to predict %NC from 3 selected CGM-derived features than 3 selected clinical phenotypes. LASSO regularization and partial least squares (PLS) with VIP scores were used to identify 4 CGM features that most contribute to the precision of %NC. Using factor analysis they identify 3 components that have CGM-related features: value (relating to the value of blood glucose), variability (relating to glucose variability), and autocorrelation (composed of the two autocorrelation features). These three groupings appeared in the 3 validation cohorts and when performing hierarchical clustering. To demonstrate how these three features change, a simulation was created to allow the user to examine these features under different conditions.

We appreciate reviewer #3 for the valuable and constructive comments on our manuscript.

The goal of this study was to identify CGM features that relate to %NC. Through multiple feature selection methods, they arrive at 3 components: value, variability, and autocorrelation. While the feature list is highly correlated, the authors take steps to ensure feature selection is robust. There is a lack of clarity of what each component (value, variability, and autocorrelation) includes as while similar CGM indices fall within each component, there appear to be some indices that appear as relevant to value in one dataset and to variability in the validation.

We appreciate the reviewer’s comment regarding the classification of CGMderived measures into the three components: value, variability, and autocorrelation. As the reviewer correctly points out, some measures may load differently between the value and variability components in different datasets. However, we believe that this variability reflects the inherent mathematical properties of these measures rather than a limitation of our study.

For example, the HBGI clusters differently across datasets due to its dependence on the number of glucose readings above a threshold. In populations where mean glucose levels are predominantly below this threshold, the HBGI is more sensitive to glucose variability (Fig. S3A). Conversely, in populations with a wider range of mean glucose levels, HBGI correlates more strongly with mean glucose levels (Fig. 3A). This context-dependent behaviour is expected given the mathematical properties of these measures and does not indicate an inconsistency in our classification approach.

Importantly, our main findings remain robust: CGM-derived measures systematically fall into three components-value, variability, and autocorrelation. Traditional CGM-derived measures primarily reflect either value or variability, and this categorization is consistently observed across datasets. While specific indices such as HBGI may shift classification depending on population characteristics, the overall structure of CGM data remains stable.

To address these considerations, we have added the following text to the Discussion section (lines 388-396):

Some indices, such as HBGI, showed variation in classification across datasets, with some populations showing higher factor loadings in the “mean” component and others in the “variance” component. This variation occurs because HBGI calculations depend on the number of glucose readings above a threshold. In populations where mean glucose levels are predominantly below this threshold, the HBGI is more sensitive to glucose variability (Fig. S5A). Conversely, in populations with a wider range of mean glucose levels, the HBGI correlates more strongly with mean glucose levels (Fig. 3A). Despite these differences, our validation analyses confirm that CGM-derived indices consistently cluster into three components: mean, variance, and autocorrelation.

We are sceptical about statements of significance without documentation of p-values.

We appreciate the reviewer’s concern regarding statistical significance and the documentation of p values.

First, given the multiple comparisons in our study, we used q values rather than p values, as shown in Figure 1D. Q values provide a more rigorous statistical framework for controlling the false discovery rate in multiple testing scenarios, thereby reducing the likelihood of false positives.

Second, our statistical reporting follows established guidelines, including those of the New England Journal of Medicine (Harrington, David, et al. “New guidelines for statistical reporting in the journal.” New England Journal of Medicine 381.3 (2019): 285-286.), which recommend that “reporting of exploratory end points should be limited to point estimates of effects with 95% confidence intervals” and that “replace p values with estimates of effects or association and 95% confidence intervals”. According to these guidelines, p values should not be reported in this type of study. We determined significance based on whether these 95% confidence intervals excluded zero - a method for determining whether an association is significantly different from zero (Tan, Sze Huey, and Say Beng Tan. "The correct interpretation of confidence intervals." Proceedings of Singapore Healthcare 19.3 (2010): 276-278.).

For the sake of transparency, we provide p values for readers who may be interested, although we emphasize that they should not be the basis for interpretation, as discussed in the referenced guidelines. Specifically, in Figure 1A-B, the p values for CGM_Mean, CGM_Std, and AC_Var were 0.02, 0.02, and <0.01, respectively, while those for FBG, HbA1c, and PG120 were 0.83,

0.91, and 0.25, respectively. In Figure 3C, the p values for factors 1–5 were 0.03, 0.03, 0.03, 0.24, and 0.87, respectively, and in Figure S8C, the p values for factors 1–3 were <0.01, <0.01, and 0.20, respectively.

We appreciate the opportunity to clarify our statistical methodology and are happy to provide additional details if needed.

While hesitations remain, the ability of these authors to find groupings of these many CGM metrics in relation to %NC is of interest. The believability of the associations is impeded by an obtuse presentation of the results with core data (i.e. correlation plots between CGM metrics and %NC) buried in the supplement while main figures contain plots of numerical estimates from models which would be more usefully presented in supplementary tables.

We appreciate the reviewer’s comment regarding the presentation of our results and recognize the importance of ensuring clarity and accessibility of the core data.

The central finding of our study is twofold: first, that the numerous CGM-derived measures can be systematically classified into three distinct components-mean, variance, and autocorrelation-and second, that each of these components is independently associated with %NC. This insight cannot be derived simply from examining scatter plots of individual correlations, which are provided in the Supplementary Figures. Instead, it emerges from our statistical analyses in the main figures, including multiple regression models that reveal the independent contributions of these components to %NC.

We acknowledge the reviewer’s concern regarding the accessibility of key data. To improve clarity, we have moved several scatter plots from the Supplementary Figures to the main figures (Fig. 1D-J) to allow readers to more directly visualize the relationships between CGM-derived measures and %NC. We believe this revision improved the transparency and readability of our results while maintaining the rigor of our analytical approach.

Given the small sample size in the primary analysis, there is a lot of modeling done with parameters estimated where simpler measures would serve and be more convincing as they require less data manipulation. A major example of this is that the pairwise correlation/covariance between CGM_mean, CGM_std, and AC_var is not shown and would be much more compelling in the claim that these are independent factors.

We appreciate the reviewer’s feedback on our statistical analysis and data presentation. The correlations between CGM_Mean, CGM_Std, and AC_Var were documented in Figure S1B. However, to improve accessibility and clarity, we have moved these correlation analyses to the main figures (Fig. 1F).

Regarding our modeling approach, we chose LASSO and PLS methods because they are wellestablished techniques that are particularly suited to scenarios with many input variables and a relatively small sample size. These methods have been used in the literature as robust approaches for variable selection under such conditions (Tibshirani R. 1996). Regression shrinkage and selection via the lasso. J R Stat Soc 58:267–288. Wold S, Sjöström M, Eriksson L. 2001. PLS-regression: a basic tool of chemometrics. Chemometrics Intellig Lab Syst 58:109–130. Pei X, Qi D, Liu J, Si H, Huang S, Zou S, Lu D, Li Z. 2023. Screening marker genes of type 2 diabetes mellitus in mouse lacrimal gland by LASSO regression. Sci Rep 13:6862. Wang C, Kong H, Guan Y, Yang J, Gu J, Yang S, Xu G. 2005. Plasma phospholipid metabolic profiling and biomarkers of type 2 diabetes mellitus based on high-performance liquid chromatography/electrospray mass spectrometry and multivariate statistical analysis.

Anal Chem 77:4108–4116.

Lack of methodological detail is another challenge. For example, the time period of CGM metrics or CGM placement in the primary study in relation to the IVUS-derived measurements of coronary plaques is unclear. Are they temporally distant or proximal/ concurrent with the PCI?

We appreciate the reviewer’s important question regarding the temporal relationship between CGM measurements and IVUS-derived plaque assessments. As described in our previous work (Otowa‐Suematsu, Natsu, et al. “Comparison of the relationship between multiple parameters of glycemic variability and coronary plaque vulnerability assessed by virtual histology–intravascular ultrasound.” Journal of Diabetes Investigation 9.3 (2018): 610615.), all individuals underwent continuous glucose monitoring for at least three consecutive days within the seven-day period prior to the PCI procedure. To improve clarity for readers, we have added the following text to the Methods section (lines 440-441):

All individuals underwent CGM for at least three consecutive days within the seven-day period prior to the PCI procedure.

A patient undergoing PCI for coronary intervention would be expected to have physiological and iatrogenic glycemic disturbances that do not reflect their baseline state. This is not considered or discussed.

We appreciate the reviewer’s concern regarding potential glycemic disturbances associated with PCI. As described in our previous work (Otowa‐Suematsu, Natsu, et al. “Comparison of the relationship between multiple parameters of glycemic variability and coronary plaque vulnerability assessed by virtual histology–intravascular ultrasound.” Journal of Diabetes Investigation 9.3 (2018): 610-615.), all CGM measurements were performed before the PCI procedure. This temporal separation ensures that the glycemic patterns analyzed in our study reflect the baseline metabolic state of the patients, rather than any physiological or iatrogenic effects of PCI. To avoid any misunderstanding, we have clarified this temporal relationship in the revised manuscript (lines 440-441):

All individuals underwent CGM for at least three consecutive days within the seven-day period prior to the PCI procedure.

The attempts at validation in external cohorts, Japanese, American, and Chinese are very poorly detailed. We could only find even an attempt to examine cardiovascular parameters in the Chinese data set but the outcome variables are unspecified with regard to what macrovascular events are included, their temporal relation to the CGM metrics, etc. Notably macrovascular event diagnoses are very different from the coronary plaque necrosis quantification. This could be a source of strength in the findings if carefully investigated and detailed but due to the lack of detail seems like an apples-to-oranges comparison.

We appreciate the reviewer’s comment regarding the validation cohorts and the need for greater clarity, particularly in the Chinese dataset. We acknowledge that our initial description lacked sufficient methodological detail, and we have expanded the Methods section to provide a more comprehensive explanation.

For the Chinese dataset, the data collection protocol was previously documented (Zhao, Qinpei, et al. “Chinese diabetes datasets for data-driven machine learning.” Scientific Data 10.1 (2023): 35.). Briefly, trained research staff used standardized questionnaires to collect demographic and clinical information, including diabetes diagnosis, treatment history, comorbidities, and medication use. Physical examinations included anthropometric measurements, and body mass index was calculated using standard protocols. CGM was performed using the FreeStyle Libre H device (Abbott Diabetes Care, UK), which records interstitial glucose levels at 15-minute intervals for up to 14 days. Laboratory measurements, including metabolic panels, lipid profiles, and renal function tests, were obtained within six months of CGM placement. While previous studies have linked necrotic core to macrovascular events (Xie, Yong, et al. “Clinical outcome of nonculprit plaque ruptures in patients with acute coronary syndrome in the PROSPECT study.” JACC: Cardiovascular Imaging 7.4 (2014): 397-405.), we acknowledge the limitations of the cardiovascular outcomes in the Chinese data set. These outcomes were extracted from medical records rather than standardized diagnostic procedures or imaging studies. To address these concerns, we have added the following text to the Methods section (lines 496-504):

The data collection protocol for the Chinese dataset was previously documented (Zhao et al., 2023). Briefly, trained research staff used standardized questionnaires to collect demographic and clinical information, including diabetes diagnosis, treatment history, comorbidities, and medication use. CGM records interstitial glucose levels at 15-minute intervals for up to 14 days. Laboratory measurements, including metabolic panels, lipid profiles, and renal function tests, were obtained within six months of CGM placement. While previous studies have linked necrotic core to macrovascular events, we acknowledge the limitations of the cardiovascular outcomes in the Chinese data set. These outcomes were extracted from medical records rather than from standardized diagnostic procedures or imaging studies.

Finally, the simulations at the end are not relevant to the main claims of the paper and we would recommend removing them for the coherence of this manuscript.

We appreciate the reviewer’s feedback regarding the relevance of the simulation component of our manuscript. The primary contribution of our study goes beyond demonstrating correlations between CGM-derived measures and %NC; it highlights three fundamental components of glycemic patterns-mean, variability, and autocorrelation-and their independent relationships with coronary plaque characteristics. The simulations are included to illustrate how glycemic patterns with identical means and variability can have different autocorrelation structures. Because temporal autocorrelation can be conceptually difficult to interpret, these visualizations were intended to provide intuitive examples for the readers.

However, we agree with the reviewer’s concern about the coherence of the manuscript. In response, we have streamlined the simulation section by removing simulations that do not directly support our primary conclusions (old version of the manuscript, lines 239-246, 502526), while retaining only those that enhance understanding of the three glycemic components. Regarding reviewer 2’s minor comment #4, we acknowledge that autocorrelation can be challenging to understand intuitively. To address this, we kept Fig. 4A with a brief description.

**Recommendations for the authors:**

**Reviewer 2# (Recommendations for the authors):**
Summary:The study by Sugimoto et. al. investigates the association between components of glucose dynamics-value, variability, and autocorrelation-and coronary plaque vulnerability (%NC) in patients with varying glucose tolerance levels. The research identifies three key factors that independently predict %NC and highlights the potential of continuous glucose monitoring (CGM)-derived indices in risk assessment for coronary artery disease (CAD). Using robust statistical methods and validation across diverse populations, the study emphasizes the limitations of conventional diagnostic markers and suggests a novel, CGMbased approach for improved predictive performance While the study demonstrates significant novelty and potential impact, several issues must be addressed by the authors.Major Comments:(1) The study demonstrates originality by introducing autocorrelation as a novel predictive factor in glucose dynamics, a perspective rarely explored in prior research. While the innovation is commendable, the biological mechanisms linking autocorrelation to plaque vulnerability remain speculative. Providing a hypothesis or potential pathways would enhance the scientific impact and practical relevance of this finding.

We appreciate the reviewer’s point about the need for a clearer biological explanation linking glucose autocorrelation to plaque vulnerability. Our previous research has shown that glucose autocorrelation reflects changes in insulin clearance (Sugimoto, Hikaru, et al. “Improved detection of decreased glucose handling capacities via continuous glucose monitoring-derived indices.” Communications Medicine 5.1 (2025): 103.). The relationship between insulin clearance and cardiovascular disease has been well documented (Randrianarisoa, Elko, et al. “Reduced insulin clearance is linked to subclinical atherosclerosis in individuals at risk for type 2 diabetes mellitus.” Scientific reports 10.1 (2020): 22453.), and the mechanisms described in this prior work may potentially explain the association between glucose autocorrelation and clinical outcomes observed in the present study. We have added the following sentences to the Discussion section (lines 341-352):

Despite increasing evidence linking glycemic variability to oxidative stress and endothelial dysfunction in T2DM complications (Ceriello et al., 2008; Monnier et al., 2008), the biological mechanisms underlying the independent predictive value of autocorrelation remain to be elucidated. Our previous work has shown that glucose autocorrelation is influenced by insulin clearance (Sugimoto et al., 2025), a process known to be associated with cardiovascular disease risk (Randrianarisoa et al., 2020). Therefore, the molecular pathways linking glucose autocorrelation to cardiovascular disease may share common mechanisms with those linking insulin clearance to cardiovascular disease. Although previous studies have primarily focused on investigating the molecular mechanisms associated with mean glucose levels and glycemic variability, our findings open new avenues for exploring the molecular basis of glucose autocorrelation, potentially revealing novel therapeutic targets for preventing diabetic complications.

(2) The inclusion of datasets from Japan, America, and China adds a valuable cross-cultural dimension to the study, showcasing its potential applicability across diverse populations. Despite the multi-regional validation, the sample size (n=270) is relatively small, especially when stratified by glucose tolerance categories. This limits the statistical power and applicability to diverse populations. A larger, multi-center cohort would strengthen conclusions.

We appreciate the reviewer’s concern regarding sample size and its potential impact on statistical power, especially when stratified by glucose tolerance levels. We fully agree that a larger sample size would increase statistical power, especially for subgroup analyses.

We would like to clarify several points regarding the statistical power and validation of our findings. Our study adheres to established methodological frameworks for sample size determination, including the guidelines outlined by Muyembe Asenahabi, Bostely, and Peters Anselemo Ikoha. “Scientific research sample size determination.” (2023). These guidelines balance the risks of inadequate sample size with the challenges of unnecessarily large samples. For our primary analysis examining the correlation between CGM-derived measures and %NC, power calculations with a type I error of 0.05, a power of 0.8, and an expected correlation coefficient of 0.4 indicated that a minimum of 47 participants was required. Our sample size of 53 exceeded this threshold and allowed us to detect statistically significant correlations, as described in the Methods section.

Furthermore, our sample size aligns with previous studies investigating the associations between glucose profiles and clinical parameters, including Torimoto, Keiichi, et al. “Relationship between fluctuations in glucose levels measured by continuous glucose monitoring and vascular endothelial dysfunction in type 2 diabetes mellitus.” Cardiovascular Diabetology 12 (2013): 1-7. (n=57), Hall, Heather, et al. “Glucotypes reveal new patterns of glucose dysregulation.” PLoS biology 16.7 (2018): e2005143. (n=57), and Metwally, Ahmed A., et al. “Prediction of metabolic subphenotypes of type 2 diabetes via continuous glucose monitoring and machine learning.” Nature Biomedical Engineering (2024): 1-18. (n=32). Moreover, to provide transparency about the precision of our estimates, we have included confidence intervals for all coefficients.

Regarding the classification of glucose dynamics components, we have conducted additional validation across diverse populations including 64 Japanese, 53 American, and 100 Chinese individuals. These validation efforts have consistently supported our identification of three independent glucose dynamics components. Furthermore, the primary objective of our study was not to assess rare events, but rather to demonstrate that glucose dynamics can be decomposed into three main factors - mean, variance and autocorrelation - whereas traditional measures have primarily captured mean and variance without adequately reflecting autocorrelation. We believe that our current sample size effectively addresses this objective.

However, we acknowledge the importance of further validation on a larger scale. To address this, we conducted a large follow-up study of over 8,000 individuals with two years of followup (Sugimoto, Hikaru, et al. “Stratification of individuals without prior diagnosis of diabetes using continuous glucose monitoring” medRxiv (2025)), which confirmed our main finding that glucose dynamics consist of mean, variance, and autocorrelation. As this large study was beyond the scope of the present manuscript due to differences in primary objectives and analytical approaches, it was not included in this paper; however, it provides further support for the clinical relevance and generalizability of our findings.

To address the sample size considerations, we have added the following sentences to the Discussion section (lines 409-414):

Although our analysis included four datasets with a total of 270 individuals, and our sample size of 53 met the required threshold based on power calculations with a type I error of 0.05, a power of 0.8, and an expected correlation coefficient of 0.4, we acknowledge that the sample size may still be considered relatively small for a comprehensive assessment of these relationships. To further validate these findings, larger prospective studies with diverse populations are needed.

(3) The study focuses on a well-characterized cohort with controlled cholesterol and blood pressure levels, reducing confounding variables. However, this stringent selection might exclude individuals with significant variability in these parameters, potentially limiting the study's applicability to broader, real-world populations. The authors should discuss how this may affect generalizability and potential bias in the results.

We appreciate the reviewer’s comment regarding the potential impact of strict participant selection criteria on the broader applicability of our findings. We acknowledge that extending validation to more diverse populations would improve the generalizability of our findings.

Our validation strategy included multiple cohorts from different regions, specifically 64 Japanese, 53 American and 100 Chinese individuals. These cohorts represent a clinically diverse population, including both healthy individuals and those with diabetes, allowing for validation across a broad spectrum of metabolic conditions. However, we recognize that further validation in additional populations and clinical settings would strengthen our conclusions. To address this, we conducted a large follow-up study of over 8,000 individuals with two years of follow-up (Sugimoto, Hikaru, et al. “Stratification of individuals without prior diagnosis of diabetes using continuous glucose monitoring” medRxiv (2025)), which confirmed our main finding that glucose dynamics consist of mean, variance, and autocorrelation. As this large study was beyond the scope of the present manuscript due to differences in primary objectives and analytical approaches, it was not included in this paper; however, it provides further support for the clinical relevance and generalizability of our findings.

We have added the following text to the Discussion section to address these considerations (lines 409-414, 354-361):

Although our analysis included four datasets with a total of 270 individuals, and our sample size of 53 met the required threshold based on power calculations with a type I error of 0.05, a power of 0.8, and an expected correlation coefficient of 0.4, we acknowledge that the sample size may still be considered relatively small for a comprehensive assessment of these relationships. To further validate these findings, larger prospective studies with diverse populations are needed.

Although our LASSO and factor analysis indicated that CGM-derived measures were strong predictors of %NC, this does not mean that other clinical parameters, such as lipids and blood pressure, are irrelevant in T2DM complications. Our study specifically focused on characterizing glucose dynamics, and we analyzed individuals with well-controlled serum cholesterol and blood pressure to reduce confounding effects. While we anticipate that inclusion of a more diverse population would not alter our primary findings regarding glucose dynamics, it is likely that a broader data set would reveal additional predictive contributions from lipid and blood pressure parameters.

(4) The study effectively highlights the potential of CGM-derived indices as a tool for CAD risk assessment, a concept that aligns with contemporary advancements in personalized medicine. Despite its potential, the complexity of CGM-derived indices like AC_Var and ADRR may hinder their routine clinical adoption. Providing simplified models or actionable guidelines would facilitate their integration into everyday practice.

We appreciate the reviewer’s concern about the complexity of CGM-derived indices such as AC_Var and ADRR for routine clinical use. We recognize that for these indices to be of practical use, they must be both interpretable and easily accessible to healthcare providers.

To address this, we have developed an easy-to-use web application that automatically calculates these measures, including AC_Var, mean glucose levels, and glucose variability. By eliminating the need for manual calculations, this tool streamlines the process and makes these indices more practical for clinical use.

Regarding interpretability, we acknowledge that establishing specific clinical guidelines would enhance the practical utility of these measures. For example, defining a cut-off value for AC_Var above which the risk of diabetes complications increases significantly would provide clearer clinical guidance. However, given our current sample size limitations and our predefined objective of investigating correlations among indices, we have taken a conservative approach by focusing on the correlation between AC_Var and %NC rather than establishing definitive cutoffs. This approach intentionally avoids problematic statistical practices like phacking. It is not realistic to expect a single study to accomplish everything from proposing a new concept to conducting large-scale clinical trials to establishing clinical guidelines. Establishing clinical guidelines typically requires the accumulation of multiple studies over many years. Recognizing this reality, we have been careful in our manuscript to make modest claims about the discovery of new “correlations” rather than exaggerated claims about immediate routine clinical use.

To address this limitation, we conducted a large follow-up study of over 8,000 individuals in the next study (Sugimoto, Hikaru, et al. “Stratification of individuals without prior diagnosis of diabetes using continuous glucose monitoring” medRxiv (2025)), which proposed clinically relevant cutoffs and reference ranges for AC_Var and other CGM-derived indices. As this large study was beyond the scope of the present manuscript due to differences in primary objectives and analytical approaches, it was not included in this paper; however, by integrating automated calculation tools with clear clinical thresholds, we expect to make these measures more accessible for clinical use.

We have added the following text to the Discussion section to address these considerations (lines 415-419):

While CGM-derived indices such as AC_Var and ADRR hold promise for CAD risk assessment, their complexity may present challenges for routine clinical implementation. To improve usability, we have developed a web-based calculator that automates these calculations. However, defining clinically relevant thresholds and reference ranges requires further validation in larger cohorts.

(5) The exclusion of TIR from the main analysis is noted, but its relevance in diabetes management warrants further exploration. Integrating TIR as an outcome measure could provide additional clinical insights.

We appreciate the reviewer’s comment regarding the potential role of time in range (TIR) as an outcome measure in our study. Because TIR is primarily influenced by the mean and variance of glucose levels, it does not fully capture the distinct role of glucose autocorrelation, which was the focus of our investigation.

To clarify this point, we have expanded the Discussion section as follows (lines 380-388):

Although time in range (TIR) was not included in the main analyses due to the relatively small number of T2DM patients and the predominance of participants with TIR >70%, our results demonstrate that CGM-derived indices outperformed conventional markers such as FBG, HbA1c, and PG120 in predicting %NC. Furthermore, multiple regression analysis between factor scores and TIR revealed that only factor 1 (mean) and factor 2 (variance) were significantly associated with TIR (Fig. S8C, D). This finding confirms the presence of three distinct components in glucose dynamics and highlights the added value of examining AC_Var as an independent glycemic feature beyond conventional CGM-derived measures.

(6) While the study reflects a commitment to understanding CAD risks in a global context by including datasets from Japan, America, and China, the authors should provide demographic details (e.g., age, gender, socioeconomic status) and discuss how these factors might influence glucose dynamics and coronary plaque vulnerability.

We appreciate the reviewer’s comment regarding the potential influence of demographic factors on glucose dynamics and coronary plaque vulnerability. We examined these relationships and found that age and sex had minimal effects on glucose dynamics characteristics, as shown in Figure S8A and S8B. These findings suggest that our primary conclusions regarding glucose dynamics and coronary risk remain robust across demographic groups within our data set.

To address the reviewer’s suggestion, we have added the following discussion (lines 361-368):

In our analysis of demographic factors, we found that age and gender had minimal influence on glucose dynamics characteristics (Fig. S8A, B), suggesting that our findings regarding the relationship between glucose dynamics and coronary risk are robust across different demographic groups within our dataset. Future studies involving larger and more diverse populations would be valuable to comprehensively elucidate the potential influence of age, gender, and other demographic factors on glucose dynamics characteristics and their relationship to cardiovascular risk.

(7) While the article shows CGM-derived indices outperform traditional markers (e.g., HbA1c, FBG, PG120), it does not compare these indices against existing advanced risk models (e.g., Framingham Risk Score for CAD). A direct comparison would strengthen the claim of superiority.

We appreciate the reviewer’s comment regarding the comparison of CGMderived indices with existing CAD risk models. Given that our study population consisted of individuals with well-controlled total cholesterol and blood pressure levels, a direct comparison with the Framingham Risk Score for Hard Coronary Heart Disease (Wilson, Peter WF, et al. “Prediction of coronary heart disease using risk factor categories.” Circulation 97.18 (1998): 1837-1847.) may introduce inherent bias, as these factors are key components of the score.

Nevertheless, to further assess the predictive value of the CGM-derived indices, we performed additional analyses using linear regression to predict %NC. Using the Framingham Risk Score, we obtained an R² of 0.04 and an Akaike Information Criterion (AIC) of 330. In contrast, our proposed model incorporating the three glycemic parameters - CGM_Mean, CGM_Std, and AC_Var - achieved a significantly improved R² of 0.36 and a lower AIC of 321, indicating superior predictive accuracy. We have updated the Result section as follows (lines 115-122):

The regression model including CGM_Mean, CGM_Std and AC_Var to predict %NC achieved an R^2^ of 0.36 and an Akaike Information Criterion (AIC) of 321. Each of these indices showed statistically significant independent positive correlations with %NC (Fig. 1A). In contrast, the model using conventional glycemic markers (FBG, HbA1c, and PG120) yielded an R² of only 0.05 and an AIC of 340 (Fig. 1B). Similarly, the model using the Framingham Risk Score for Hard Coronary Heart Disease (Wilson et al., 1998) showed limited predictive value, with an R² of 0.04 and an AIC of 330 (Fig. 1C).

(8) The study mentions varying CGM sampling intervals across datasets (5-minute vs. 15minute). Authors should employ sensitivity analysis to assess the impact of these differences on the results. This would help clarify whether higher-resolution data significantly improves predictive performance.

We appreciate the reviewer’s comment regarding the potential impact of different CGM sampling intervals on our results. To assess the robustness of our findings across different sampling frequencies, we performed a down sampling analysis by converting our 5minute interval data to 15-minute intervals. The AC_Var value calculated from 15-minute intervals was significantly correlated with that calculated from 5-minute intervals (R = 0.99, 95% CI: 0.97-1.00). Consequently, the main findings remained consistent across both sampling frequencies, indicating that our results are robust to variations in temporal resolution. We have added this analysis to the Result section (lines 122-126):

The AC_Var computed from 15-minute CGM sampling was nearly identical to that computed from 5-minute sampling (R = 0.99, 95% CI: 0.97-1.00) (Fig. S1A), and the regression using the 15‑min features yielded almost the same performance (R^2^ = 0.36; AIC = 321; Fig. S1B).

(9) The identification of actionable components in glucose dynamics lays the groundwork for clinical stratification. The authors could explore the use of CGM-derived indices to develop a simple framework for stratifying risk into certain categories (e.g., low, moderate, high). This could improve clinical relevance and utility for healthcare providers.

We appreciate the reviewer’s suggestion regarding the potential for CGMderived indices to support clinical stratification. We completely agree with the idea that establishing risk categories (e.g., low, moderate, high) based on specific thresholds would enhance the clinical utility of these measures. However, given our current sample size limitations and our predefined objective of investigating correlations among indices, we have taken a conservative approach by focusing on the correlation between AC_Var and %NC rather than establishing definitive cutoffs. This approach intentionally avoids problematic statistical practices like p-hacking. It is not realistic to expect a single study to accomplish everything from proposing a new concept to conducting large-scale clinical trials to establishing clinical thresholds. Establishing clinical thresholds typically requires the accumulation of multiple studies over many years. Recognizing this reality, we have been careful in our manuscript to make modest claims about the discovery of new “correlations” rather than exaggerated claims about immediate routine clinical use.

To address this limitation, we conducted a large follow-up study of over 8,000 individuals in the next study (Sugimoto, Hikaru, et al. “Stratification of individuals without prior diagnosis of diabetes using continuous glucose monitoring” medRxiv (2025)), which proposed clinically relevant cutoffs and reference ranges for AC_Var and other CGM-derived indices. As this large study was beyond the scope of the present manuscript due to differences in primary objectives and analytical approaches, it was not included in this paper. However, we expect to make these measures more actionable in clinical use by integrating automated calculation tools with clear clinical thresholds.

We have added the following text to the Discussion section to address these considerations (lines 415-419):

While CGM-derived indices such as AC_Var and ADRR hold promise for CAD risk assessment, their complexity may present challenges for routine clinical implementation. To improve usability, we have developed a web-based calculator that automates these calculations. However, defining clinically relevant thresholds and reference ranges requires further validation in larger cohorts.

(10) While the study acknowledges several limitations, authors should also consider explicitly addressing the potential impact of inter-individual variability in glucose metabolism (e.g., age-related changes, hormonal influences) on the findings.

We appreciate the reviewer’s comment regarding the potential impact of interindividual variability in glucose metabolism, including age-related changes and hormonal influences, on our results. In our analysis, we found that age had minimal effects on glucose dynamics characteristics, as shown in Figure S8A. In addition, CGM-derived measures such as ADRR and AC_Var significantly contributed to the prediction of %NC independent of insulin secretion (I.I.) and insulin sensitivity (Composite index) (Fig. 2). These results suggest that our primary conclusions regarding glucose dynamics and coronary risk remain robust despite individual differences in glucose metabolism.

To address the reviewer’s suggestion, we have added the following discussion (lines 186-188, 361-368):

Conventional indices, including FBG, HbA1c, PG120, I.I., Composite index, and Oral DI, did not contribute significantly to the prediction compared to these CGM-derived indices.

In our analysis of demographic factors, we found that age and gender had minimal influence on glucose dynamics characteristics (Fig. S8A, B), suggesting that our findings regarding the relationship between glucose dynamics and coronary risk are robust across different demographic groups within our dataset. Future studies involving larger and more diverse populations would be valuable to comprehensively elucidate the potential influence of age, gender, and other demographic factors on glucose dynamics characteristics and their relationship to cardiovascular risk.

(11) It's unclear whether the identified components (value, variability, and autocorrelation) could serve as proxies for underlying physiological mechanisms, such as beta-cell dysfunction or insulin resistance. Please clarify.

We appreciate the reviewer’s comment regarding the physiological underpinnings of the glucose components we identified. The mean, variance, and autocorrelation components we identified likely reflect specific underlying physiological mechanisms related to glucose regulation. In our previous research (Sugimoto, Hikaru, et al. “Improved detection of decreased glucose handling capacities via continuous glucose monitoring-derived indices.” Communications Medicine 5.1 (2025): 103.), we explored the relationship between glucose dynamics characteristics and glucose control capabilities using clamp tests and mathematical modelling. These investigations revealed that autocorrelation specifically shows a significant correlation with the disposition index (the product of insulin sensitivity and insulin secretion) and insulin clearance parameters.

Furthermore, our current study demonstrates that CGM-derived measures such as ADRR and AC_Var significantly contributed to the prediction of %NC independent of established metabolic parameters including insulin secretion (I.I.) and insulin sensitivity (Composite index), as shown in Figure 2. These results suggest that the components we identified capture distinct physiological aspects of glucose metabolism beyond traditional measures of beta-cell function and insulin sensitivity. Further research is needed to fully characterize these relationships, but our results imply that these characteristics of glucose dynamics offer supplementary insight into the underlying beta-cell dysregulation that contributes to coronary plaque vulnerability.

To address the reviewer’s suggestion, we have added the following discussion to the Result section (lines 186-188):

Conventional indices, including FBG, HbA1c, PG120, I.I., Composite index, and Oral DI, did not contribute significantly to the prediction compared to these CGM-derived indices.

Minor Comments:(1) The use of LASSO and PLS regression is appropriate, but the rationale for choosing these methods over others (e.g., Ridge regression) should be explained in greater detail.

We appreciate the reviewer’s comment and have added the following discussion to the Methods section (lines 578-585):

LASSO regression was chosen for its ability to perform feature selection by identifying the most relevant predictors. Unlike Ridge regression, which simply shrinks coefficients toward zero without reaching exactly zero, LASSO produces sparse models, which is consistent with our goal of identifying the most critical features of glucose dynamics associated with coronary plaque vulnerability. In addition, we implemented PLS regression as a complementary approach due to its effectiveness in dealing with multicollinearity, which was particularly relevant given the high correlation among several CGM-derived measures.

(2) While figures are well-designed, adding annotations to highlight key findings (e.g., significant contributors in factor analysis) would improve clarity.

We appreciate the reviewer’s suggestion to improve the clarity of our figures. In the factor analysis, we decided not to include annotations because indicators such as ADRR and J-index can be associated with multiple factors, which could lead to misleading or confusing interpretations. However, in response to the suggestion, we have added annotations to the PLS analysis, specifically highlighting items with VIP values greater than 1 (Fig. 2D, S2D) to emphasize key contributors.

(3) The term "value" as a component of glucose dynamics could be clarified. For instance, does it strictly refer to mean glucose levels, or does it encompass other measures?

We appreciate the reviewer’s question regarding the term “value” in the context of glucose dynamics. Factor 1 was predominantly influenced by CGM_Mean, with a factor loading of 0.99, indicating that it primarily represents mean glucose levels. Given this strong correlation, we have renamed Factor 1 to “Mean” (Fig. 3A) to more accurately reflect its role in glucose dynamics.

(4) The concept of autocorrelation may be unfamiliar to some readers. A brief, intuitive explanation with a concrete example of how it manifests in glucose dynamics would enhance understanding.

We appreciate the reviewer’s suggestion. Autocorrelation refers to the relationship between a variable and its past values over time. In the context of glucose dynamics, it reflects how current glucose levels are influenced by past levels, capturing patterns such as sustained hyperglycemia or recurrent fluctuations. For example, if an individual experiences sustained high glucose levels after a meal, the strong correlation between successive glucose readings indicates high autocorrelation. We have included this explanation in the revised manuscript (lines 519-524) to improve clarity for readers unfamiliar with the concept. Additionally, Figure 4A shows an example of glucose dynamics with different autocorrelation.

(5) Ensure consistent use of terms like "glucose dynamics," "CGM-derived indices," and "plaque vulnerability." For instance, sometimes indices are referred to as "components," which might confuse readers unfamiliar with the field.

We appreciate the reviewer’s comment about ensuring consistency in terminology. To avoid confusion, we have reviewed and standardized the use of terms such as “CGM-derived indices,” and “plaque vulnerability” throughout the manuscript. Additionally, while many of our measures are strictly CGM-derived indices, several “components” in our analysis include fasting blood glucose (FBG) and glucose waveforms during the OGTT. For these measures, we retained the descriptors “glucose dynamics” and “components” rather than relabelling them as CGM-derived indices.

(6) Provide a more detailed overview of the supplementary materials in the main text, highlighting their relevance to the key findings.

We appreciate the reviewer’s suggestion. We revised the manuscript by integrating the supplementary text into the main text (lines 129-160), which provides a clearer overview of the supplementary materials. Consequently, the Supplementary Information section now only contains supplementary figures, while their relevance and key details are described in the main text.

**Reviewer #3 (Recommendations for the authors):**
Other Concerns:(1) The text states the significance of tests, however, no p-values are listed: Lines 118-119: Significance is cited between CGM indices and %NC, however, neither the text nor supplementary text have p-values. Need p-values for Figure 3C, Figure S10. When running the https://cgm-basedregression.streamlit.app/ multiple regression analysis, a p-value should be given as well. Do the VIP scores (Line 142) change with the inclusion of SBP, DBP, TG, LDL, and HDL? Do the other datasets have the same well-controlled serum cholesterol and BP levels?

We appreciate the reviewer’s concern regarding statistical significance and the documentation of p values.

First, given the multiple comparisons in our study, we used q values rather than p values, as shown in Figure 1D. Q values provide a more rigorous statistical framework for controlling the false discovery rate in multiple testing scenarios, thereby reducing the likelihood of false positives.

Second, our statistical reporting follows established guidelines, including those of the New England Journal of Medicine (Harrington, David, et al. “New guidelines for statistical reporting in the journal.” New England Journal of Medicine 381.3 (2019): 285-286.), which recommend that “reporting of exploratory end points should be limited to point estimates of effects with 95% confidence intervals” and that “replace p values with estimates of effects or association and 95% confidence intervals”. According to these guidelines, p values should not be reported in this type of study. We determined significance based on whether these 95% confidence intervals excluded zero - a statistical method for determining whether an association is significantly different from zero (Tan, Sze Huey, and Say Beng Tan. “The correct interpretation of confidence intervals.” Proceedings of Singapore Healthcare 19.3 (2010): 276-278.).

For the sake of transparency, we provide p values for readers who may be interested, although we emphasize that they should not be the basis for interpretation, as discussed in the referenced guidelines. Specifically, in Figure 1A-B, the p values for CGM_Mean, CGM_Std, and AC_Var were 0.02, 0.02, and <0.01, respectively, while those for FBG, HbA1c, and PG120 were 0.83, 0.91, and 0.25, respectively. In Figure 3C, the p values for factors 1–5 were 0.03, 0.03, 0.03, 0.24, and 0.87, respectively, and in Figure S8C, the p values for factors 1–3 were <0.01, <0.01, and 0.20, respectively. We appreciate the opportunity to clarify our statistical methodology and are happy to provide additional details if needed.

We confirmed that the results of the variable importance in projection (VIP) analysis remained stable after including additional covariates, such as systolic blood pressure (SBP), diastolic blood pressure (DBP), triglycerides (TG), low-density lipoprotein cholesterol (LDL-C), and high-density lipoprotein cholesterol (HDL-C). The VIP values for ADRR, MAGE, AC_Var, and LI consistently exceeded one even after these adjustments, suggesting that the primary findings are robust in the presence of these clinical variables. We have added the following sentences in the Results and Methods section (lines 188-191, 491-494):

Even when SBP, DBP, TG, LDL-C, and HDL-C were included as additional input variables, the results remained consistent, and the VIP scores for ADRR, AC_Var, MAGE, and LI remained greater than 1 (Fig. S2D).

Of note, as the original reports document, the validation datasets did not specify explicit cutoffs for blood pressure or cholesterol. Consequently, they included participants with suboptimal control of these parameters.

(2) Negative factor loadings have not been addressed and consistency in components: Figure 3, Figure S7. All the main features for value in Figure 3A are positive. However, MVALUE in S7B is very negative for value whereas the other features highlighted for value are positive. What is driving this difference? Please explain if the direction is important. Line 480 states that variables with factor loadings >= 0.30 were used for interpretation, but it appears in the text (Line 156, Figure 3) that oral DI was used for value, even though it had a -0.61 loading. Figure 3, Figure S7. HBGI falls within two separate components (value and variability). There is not a consistent component grouping. Removal of MAG (Line 185) and only MAG does not seem scientific. Did the removal of other features also result in similar or different Cronbach's ⍺? It is unclear what Figure S8B is plotting. What does each point mean?

We appreciate the reviewer’s comment regarding the classification of CGMderived measures into the three components: value, variability, and autocorrelation. As the reviewer correctly points out, some measures may load differently between the value and variability components in different datasets. However, we believe that this variability reflects the inherent mathematical properties of these measures rather than a limitation of our study.

For example, the HBGI clusters differently across datasets due to its dependence on the number of glucose readings above a threshold. In populations where mean glucose levels are predominantly below this threshold, the HBGI is more sensitive to glucose variability (Fig. S3A). Conversely, in populations with a wider range of mean glucose levels, HBGI correlates more strongly with mean glucose levels (Fig. 3A). This context-dependent behaviour is expected given the mathematical properties of these measures and does not indicate an inconsistency in our classification approach.

Importantly, our main findings remain robust: CGM-derived measures systematically fall into three components-value, variability, and autocorrelation. Traditional CGM-derived measures primarily reflect either value or variability, and this categorization is consistently observed across datasets. While specific indices such as HBGI may shift classification depending on population characteristics, the overall structure of CGM data remains stable.

With respect to negative factor loadings, we agree that they may appear confusing at first. However, in the context of exploratory factor analysis, the magnitude, or absolute value, of the loading is most critical for interpretation, rather than its sign. Following established practice, we considered variables with absolute loadings of at least 0.30 to be meaningful contributors to a given component. Accordingly, although the oral DI had a negative loading of –0.61, its absolute magnitude exceeded the threshold of 0.30, so it was considered in our interpretation of the “value” component. Regarding the reviewer’s observation that MVALUE in Figure S7B shows a strongly negative loading while other indices in the same component show positive loadings, we believe this reflects the relative orientation of the factor solution rather than a substantive difference in interpretation. In factor analysis, the direction of factor loadings is arbitrary: multiplying all the loadings for a given factor by –1 would not change the factor’s statistical identity. Therefore, the important factor is not whether a variable loads positively or negatively but rather the strength of its association with the latent component (i.e., the absolute value of the loading).

The rationale for removing MAG was based on statistical and methodological considerations. As is common practice in reliability analyses, we examined whether Cronbach’s α would improve if we excluded items with low factor loadings or weak item–total correlations. In the present study, we recalculated Cronbach’s α after removing the MAG item because it had a low loading. Its exclusion did not substantially affect the theoretical interpretation of the factor, which we conceptualize as “secretion” (without CGM). MAG’s removal alone is scientifically justified because it was the only item whose exclusion improved Cronbach's α while preserving interpretability. In contrast, removing other items would have undermined the conceptual clarity of the factor or would not have meaningfully improved α. Furthermore, the MAG item has a high factor 2 loading.

Each point in Figure S8B (old version) corresponds to an individual participant.

To address these considerations, we have added the following text to the Discussion, Methods, (lines 388-396, 600-601) and Figure S6B (current version) legend:

Some indices, such as HBGI, showed variation in classification across datasets, with some populations showing higher factor loadings in the “mean” component and others in the “variance” component. This variation occurs because HBGI calculations depend on the number of glucose readings above a threshold. In populations where mean glucose levels are predominantly below this threshold, the HBGI is more sensitive to glucose variability (Fig. S5A). Conversely, in populations with a wider range of mean glucose levels, the HBGI correlates more strongly with mean glucose levels (Fig. 3A). Despite these differences, our validation analyses confirm that CGM-derived indices consistently cluster into three components: mean, variance, and autocorrelation.

Variables with absolute factor loadings of ≥ 0.30 were used in interpretation.

Box plots comparing factors 1 (Mean), 2 (Variance), and 3 (Autocorrelation) between individuals without (-) and with (+) diabetic macrovascular complications. Each point corresponds to an individual. The boxes represent the interquartile range, with the median shown as a horizontal line. Mann–Whitney U tests were used to assess differences between groups, with P values < 0.05 considered statistically significant.

Minor Concerns:(1) NGT is not defined.

We appreciate the reviewer for pointing out that the term “NGT” was not clearly defined in the original manuscript. We have added the following text to the Methods section (lines 447-451):

T2DM was defined as HbA1c ≥ 6.5%, fasting plasma glucose (FPG) ≥ 126 mg/dL or 2‑h plasma glucose during a 75‑g OGTT (PG120) ≥ 200 mg/dL. IGT was defined as HbA1c 6.0– 6.4%, FPG 110–125 mg/dL or PG120 140–199 mg/dL. NGT was defined as values below all prediabetes thresholds (HbA1c < 6.0%, FPG < 110 mg/dL and PG120 < 140 mg/dL).

(2) Is it necessary to list the cumulative percentage (Line 173), it could be clearer to list the percentage explained by each factor instead.

We appreciate the reviewer’s suggestion to list the percentage explained by each factor rather than the cumulative percentage for improved clarity. According to the reviewer’s suggestion, we have revised the results to show the individual contribution of each factor (39%, 21%, 10%, 5%, 5%) rather than the cumulative percentages (39%, 60%, 70%, 75%, 80%) that were previously listed (lines 220-221).

(3) Figure S10. How were the coefficients generated for Figure S10? No methods are given.

We conducted a multiple linear regression analysis in which time in range (TIR) was the dependent variable and the factor scores corresponding to the first three latent components (factor 1 representing the mean, factor 2 representing the variance, and factor 3 representing the autocorrelation) were the independent variables. We have added the following text to the figure legend (Fig. S8C) to provide a more detailed description of how the coefficients were generated:

Comparison of predicted Time in range (TIR) versus measured TIR using multiple regression analysis between TIR and factor scores in Figure 3. In this analysis, TIR was the dependent variable, and the factor scores corresponding to the first three latent components (factor 1 representing the mean, factor 2 representing the variance, and factor 3 representing the autocorrelation) were the independent variables. Each point corresponds to the values for a single individual.

(4) In https://cgm-basedregression.streamlit.app/, more explanation should be given about the output of the multiple regression. Regression is spelled incorrectly on the app**.**

We appreciate the reviewer for pointing out the need for a clearer explanation of the multiple regression analysis presented in the online tool

(https://cgmregressionapp2.streamlit.app/). We have added the description about the regression and corrected the typographical error in the spelling of “regression” within the app.

(5) The last section of results (starting at line 225) appears to be unrelated to the goal of predicting %NC.

We appreciate the reviewer’s feedback regarding the relevance of the simulation component of our manuscript. The primary contribution of our study goes beyond demonstrating correlations between CGM-derived measures and %NC; it highlights three fundamental components of glycemic patterns-mean, variance, and autocorrelation-and their independent relationships with coronary plaque characteristics. The simulations are included to illustrate how glycemic patterns with identical means and variability can have different autocorrelation structures. As reviewer 2 pointed out in minor comment #4, temporal autocorrelation can be difficult to interpret, so these visualizations were intended to provide intuitive examples for readers.

However, we agree with the reviewer’s concern about the coherence of the manuscript. In response, we have streamlined the simulation section by removing technical simulations that do not directly support our primary conclusions (old version of the manuscript, lines 239-246, 502-526), while retaining only those that enhance understanding of the three glycemic components (Fig. 4A).

(6) Figure S2. The R2 should be reported.

We appreciate the reviewer for suggesting that we report R² in Figure S2. In the revised version, we have added the correlation coefficients and their 95% confidence intervals to Figure 1E.

(7) Multiple panels have a correlation line drawn with a slope of 1 which does not reflect the data or r^2 listed. this should be fixed.

We appreciate the reviewer’s concern that several panels included regression lines with a fixed slope of one that did not reflect the associated R² values. We have corrected Figures 1A–C and 3C to display regression lines representing the estimated slopes derived from the regression analyses.